



# Snow depth estimation by time-lapse photography: Finnish and Italian case studies

Marco Bongio [1], Ali Nadir Arslan[2], Cemal Melih Tanis[3] Carlo De Michele[1]

[1] Department of Civil and Environmental Engineering, Politecnico di Milano, Piazza Leonardo da Vinci 32, 20133, Milano, Italy

[2] Finnish Meteorological Institute, Erik Palménin aukio 1, P.O.Box 503, FI-00101 Helsinki, Finland

[3] Poste Restante 00100, Helsinki, Finland

*Correspondence to*: Marco Bongio (marco.bongio@polimi.it)

**Abstract.** We explored the potentiality of time-lapse photography method to estimate the snow depth in boreal forested and alpine regions. Historically, the snow depth has been measured manually by rulers or snowboards, with a temporal resolution of once per day, and a time-consuming activity. In the last decades, ultrasonic and/or optical sensors have been developed to obtain automatic measurements with higher temporal resolution and accuracy, defining a network of sensors within each country. The Finnish Meteorological Institute Image processing tool (FMIPROT) is used to retrieve the snow depth from images of a snow stake on the ground collected by cameras. An "ad-hoc" algorithm based on the brightness difference between snowpack and stake's markers has been developed. We illustrated three case studies to highlight potentialities and pitfalls of the method. The proposed method provides new possibilities and advantages in the estimation of snow depth, which can be summarized as follows: 1) retrieving the snow depth at high temporal resolution with an accuracy comparable to the most common method (manual measurements); 2) visual identification of errors or misclassifications; 3) estimating the spatial variability of snow depth; 4) correction the well-known under catch problem of instrumental pluviometer; 5) retrieval of snow depth in avalanche, dangerous and inaccessible sites, where there is in general a lack of data; 6) cheap, reliable, flexible and easily extendible in different environments and applications. Root Mean Square Errors (RMSE) and Nash Sutcliffe Efficiency (NSE) are calculated for all three case studies comparing with estimates from both the FMIPROT and visual inspection of images. For the case studies, $NSE = 0.917$, $0.963$, $0.916$ were respectively for Sodankylä, Gressoney and Careser. In terms of accuracy, the Sodankylä case study gave better results (RMSE equal to $3.951 \cdot 10^{-2}m$, $5.242 \cdot 10^{-2}m$, $10.78 \cdot 10^{-2}m$, respectively). The worst performances occurred at Careser dam located at 2600 m a.s.l. where extreme weather conditions occur, strongly affecting the clarity of the images.





# 1. INTRODUCTION

### 1.1 Snow water equivalent and snow depth

Seasonal snow is an important part of the Earth's climate system and has a strong influence on the Earth Energy balance. Due to the inhomogeneous spatial distribution of snow, traditional in-situ measurement techniques can hardly provide exhaustive information about snow variability (Lundberg et al. 2010). Remote sensing is becoming the most widespread technique to evaluate the snow cover at large scales (Takala et al. 2011).

From the hydrological point of view, the main variable of snowpack is the snow water equivalent (SWE), rather than snow

depth (SD), or snow density ($\rho_S$) (Avanzi et. al, 2015). But most of the historical information related to the snow are the manual measurements of snow depth collected by rulers or snowboards. If the snow depth can be easily retrieved with such instruments having enough accuracy, the conversion between snow depth and snow water equivalent is a delicate issue due to the temporal variability of the snow density influenced by a lot of factors such as atmospheric conditions and topographic effects. The snow water equivalent is a function of snow depth and snow density as $SWE(t) = SD(t)\frac{\rho_S(t)}{\rho_w}$ (De Walle and Rango, 2008) where

$\rho_w$ is the density of water. Thus, the uncertainty associated with the snow water equivalent estimation is a function of both snow depth and snow density estimation. Uncertainties related to snow depth measurements depend on the temporal and spatial resolution of the measurement methods and will be discuss in the next section.

About the fresh snow density ($\rho_{FS}$) estimation, there are a lot of studies where the density is expressed as a function of some meteorological parameters such as air temperature (T), wind speed (u), and relative humidity (RH) (Bavera et al. 2012) and

can be classified as site-specific and empirical because they were determined after a procedure of best fitting based on the comparison between model and measurements.

If the fresh snow density ($\rho_{FS}$) is difficult to define, the density of snowpack ($\rho_S$) is generally characterized by much more complexity, due to concurrent processes of snow accumulation and modification. To estimate $\rho_S$ two approaches can be identified: the first and simplest one is to find empirical relationships, between density and other meteorological variables such

as: temperature, wind speed, days from the conventional date , (Bavera and De Michele 2009, Jonas et al. 2009; Bavera et al. 2012; Avanzi et al. 2015, Helfricht et al. 2018). The second one is to reproduce adequately snowpack dynamics with mathematical models driven with temperature, precipitation, wind measurements which considered all the physical processes which affected the snowpack evolution The most used are: SNOWPACK (Bartlet et al. 2002; Lehning et al. 2002; Wever et al. 2014); CROCUS (Brun et al. 1992; Carmagnola et al. 2014;Vionnet et al. 2013); HyS (De Michele et al. 2013, Avanzi et

al. 2014; 2015). But these models can be defined as point models, because they were defined to calculate the snowpack dynamics only in one specific site or at maximum at watershed level. Moreover, these models are strongly dependent of the accuracy of the inputs: precipitation, temperature and wind quantification. If the last two can be easily captured and corrected, the first one, in some cases, could be very challenging. In fact, precipitation underestimation by pluviometers can reach values from 50% to 60% of the total volume especially when the ratio between snowfall and rainfall is high (mountain regions). This

is due to many factors: wind field deformation above the gauge orifice, wetting loss on the internal walls of the collector, wetting loss in the container when it's emptied, evaporation from the container, blowing and drifting snow, in and out splashing of water (Goodison 1998; Legates and Willmott 1990). These errors generally increase with altitude because much more precipitation fall as snow, and steep and mountain regions are characterized by high temperature and pressure gradients which can generate extreme windy conditions. Some authors tried to define relationships to correct pluviometer under catch, defining

those with the use of some hydrological variables as predictors (temperature, wind, humidity) (Korchendorfer et al. 2017; Sevruk 1987). With the help of time-lapse digital imagery processing, we showed that we can define a reliable snow depth time series. Differentiating the time series at each time step we can define the snowfall rate, considering the positive difference between two subsequent values. Measuring directly the accumulated snowpack, the pluviometer undercatch problem can be avoided. The proposed method does not consider snow settling, but with hourly time resolution this process can be overlooked.



### 1.2 Traditional methods to measure snow depth

There are many different techniques for estimating the snow on the ground (MSG), which can be categorized into three main groups: from Airborne Measurements (AM), from Spaceborne (S) and Ground Measurements (GM). Each of these is characterized by its own temporal and spatial resolution.

About AM, there are two principal methods: photogrammetry and LIDAR-based approaches. The first one includes determining the geometric properties of objects through the identification of common points within photographic images retrieved from different positions. However, in the last decade with proficiency results, UAS (Unmanned Aerial System) technology has been developed permitting flexible, efficient and economic data acquisitions, even within inaccessible alpine terrains (De Michele et al. 2016, Avanzi et al. 2018, Bühler et al. 2016, Vander Jagt et al. 2015).

LIDAR is an active ranging instrument, which emits laser pulses at specified frequency levels, and measures the flight time between the transmitted and return signal (Tedesco 2014). The accuracy of the snow depth retrieval depends on many parameters: pulse repetition frequency, scan rate and angle, beam divergence, scan pattern, along and across track point spacing, swath width. This method could have a great potential especially in avalanche science application and inaccessible and very steep sites, obtaining a snow depth estimation with high accuracy at sub-basin level, but nowadays is just too expensive for extensive applications.

The Spaceborne (S) measurements can be conducted with passive and active sensors. In literature it can be found two different techniques which use the passive sensor, the first one based on brightness temperature measurements and the second one focused on the signal's phase (Che et al. 2008). These methods do not allow to measure the snow depth at high resolution (less than $0.2\ m$), but they are commonly used to identify the fractional snow cover within a regional or continental scale depending on the satellite's swath (Qiao et al. 2018).

Active sensors emit an electromagnetic signal and collected the response which can be reflected and/or transmitted by the soil and snow layer. The transmitted energy is attenuated by the snowpack, which can be seen as the sum of many contributions: from the air-snow interface, the volumetric one related to liquid water content, density, snow grain size and the reflected energy from the ground-snow interface (Azar et al. 2006; Papa et al. 2002). Also, in this case, accuracy of snow depth estimation is not so high (less than $0.2\ m$). In the last decade, methods which combine passive and active microwave signals to estimate the snow depth with high accuracy have been developed (Liu et al. 2017).

On the contrary of the airborne and spaceborne measurements, recently developed, ground measurements of snow have a very long history. The first historical measurements were conducted with rulers or snowboards, generally with daily temporal resolution. After 90' automatic weather stations were equipped with ultrasonic snow depth sensors and the opto-electronic ones which can improve the accuracy within 1 mm have been developing. Recently, disdrometers are used to classify precipitation as liquid or solid based on the particle radius and velocity.

If manual measurements are characterized by low temporal resolution but high reliability, generally ultrasonic and optical sensors give very noisy snow depth time series but with very high temporal resolution (Pirazzini et al. 2018).

Accuracy, temporal and spatial resolution and applications of existing methods in snow depth retrieval are reported in Table S1: there is not any method which can be considered reliable under all points of view, not due to the measurement uncertainties, or lack of data, but due to the difficulties which characterize the dynamics of the snowpack. In addition, each method and the results reliability depend strongly to the location of case study. In fact, in high mountain terrain or avalanche region, or where climatic conditions are worst, each method cannot retrieve alone the right snow depth value. From these reasons emerge the need of another method, which is cheap and reliable at the same time.

### 1.3 Application of time-lapse photography to cryosphere monitoring

In literature the time-lapse photography is used to determine some characteristics of the snow such as: snow depth, snow canopy interception, snow settling, fractional snow cover on the ground, albedo, state of precipitation. Mainly applications in





remote areas, Greenland or Arctic Region, or in mountain regions, were related to study snow accumulation on glacierized areas (Christiansen 2001, Floyd and Weiler 2008, Farinotti et al. 2010; Parajka et al. 2012; Bernard et al. 2013; Garvelmann et al. 2013; Hedrick and Marshall 2014; Dong and Menzel 2017). In these studies, the principal aim is to show how time-lapse photography could be used for investigating snow processes, but they were not properly focused in snow depth retrieval purposes.

Starting from 2001, continuous automatic digital photography was tested in high-Arctic Greenland to monitor snow-cover conditions by Christiansen (2001). Daily photographs covered a 100 m transect through a seasonal snow patch, and thus on an annual basis also yields information on snow-cover duration in the different vegetation zones of the snow patch. Photographs, combined with measurements taken by Automatic Weather Stations, allowed to study snow wind induced redistribution on different thin snow-covered areas. Christiansen (2001) suggested that this method can be seen as an alternative to the traditional snow monitoring methods providing areal information and not only point measurements. Floyd and Weiler (2008) designed an automatic time-lapse photography network to monitor the state of precipitation (rain vs. snow), snow accumulation/ablation, canopy interception and unloading of snow from the canopy; they also defined an image analysis software which can calculate snow parameters from images.

Farinotti et al. (2010) tried to use conventional oblique photography combined with a temperature index melt and accumulation model to infer the snow accumulation distribution of a small Alpine catchment. The inferred snow accumulation distribution was validated with in-situ measurements and correlations with topographic variables, such as curvature and slope, were presented. The focus was on a better representation of SWE and its spatial variability within a distributed hydrological model in mountain complex terrain and time-lapse photography was used to improve results.

Parajka et al. (2012) studied the potentiality of time-lapse photography of snow for hydrological purposes at small catchment scale. They designed and tested a monitoring system, which allowed multi-resolution observations of snow cover characteristics. Investigations of snow cover, snow depth, snowfall interception were carried out both in the close area near the camera and far range. Using five stakes, it was possible to crosscheck snow depth estimations, omitting the largest and smallest readings, obtaining a robust estimate of the snow depth.

Bernard et al. (2013) installed, around the Austre Lovenbreen glacier basin (Norway, 79°N), automated digital cameras in the context of long-term monitoring snow and ice dynamics with high temporal and spatial resolution. Moreover, six camera stations were oriented to observe 96% of the glacierized area.

Garvelmann et al. (2013) defined a camera network to quantify snow processes such as: snow depth, snow canopy interception, albedo and the state of precipitation. About snow depth estimations, they developed a semi-automatic approach, using images in which snow stakes with 10 cm red and black markers were visible. The main limitation was that they must define the number of pixels of a 10 cm bar of the stake and the whole stake length, with a cursor clicks procedure which makes this procedure time-consuming. In addition, the accuracy was limited by the 10 cm snow stake markers, and results comparison were made via visual inspection of images. They reported an accuracy of 7.1 cm within the period December 2011 to April 2012.

Hedrick et al. (2014), within a test site, defined an automated, low-cost and safe snow depth measurement system in avalanche terrain using time-lapse photography. They positioned two stakes painted with red color within the camera's view. They developed an algorithm that used pixel color intensities to automatically locate the snow surface for both the two stakes. The pixel counting algorithm clips each image to small rectangle around the last known vertical pixel location of the marker base, then separates and smooths the blue channel of the RGB image for ultimate considerations. This technique calculated a row wise RGB color minimum and the differences between two subsequent values. Snow surface was positioned near the maximum changing in the reflectivity level. Finally, the depth conversion was performed by subtracting the pixel row from the snow free image, then dividing by the number of pixels per centimeter for the particular marker. The snow depth estimation was compared with ultrasonic snow depth sensor and one LIDAR survey, showing a high underestimation of the implemented algorithm for the period between December 2012 and April 2013.



Dong et al. (2017) focused on snow process monitoring in mountain forests with time-lapse photography. They developed a semi-automatic procedure to interpret snow depth from digital images, which exhibited high consistency with manual measurements and station-based recordings. To estimate the snow depth, they used images in which a stake with 10 cm red and black markers was visible. The procedure required 3 different software Photoshop, ArcGIS and Excel. In particular they

calculated the maximum brightness of the top 80 cm of the snow stake and adding a fixed value (20) they defined the threshold to detect the snow surface. A pixel was assumed to represent the snow surface when the brightness values below this pixel was higher than the previous mentioned threshold. The regression model of the automatically and manually interpreted snow depth values from the digital pictures were reported, which showed that the RMSE ranged from 1.14 to 1.95 cm. Researchers should manually validate some abnormal values in accordance with the digital images obtained.

There is still lack of a full completed method to automatically estimate snow depth in real-time using snow stake images. In this study we present the FMIPROT, showing its potential, and testing it not only in a well-designed test site, but also using camera images not specifically targeted for snow depth retrieval by digital images. Innovative aspects of this work are:

- Presenting a complete and automatic software tool to estimate the snow depth from digital images, also in real-time;
- Testing the method in different sites (Boreal forest in arctic region and mountain area in Italian Alps) with camera
and stakes not properly defined for this purpose certifying the highest reliability in every condition;
- Using a temporal resolution of 1 hour and obtaining a snow depth time series for more than 1 hydrological year;
- Showing how the accuracy of snow depth estimations can be increased using stakes with 1 cm spacing markers;
- Certifying the algorithm capability, comparing estimations and visual observations, and calculating accuracy and efficiency parameters (RMSE and NSE);
- Defining the proper geometric and parameter configurations of the camera-stake system;
- Identifying sources of errors and designing post processing procedure corrections;

Finally, presented methodology can be taken as a reference for: (a) validating existing methods in Table S1, (b) comparing results in terms of amount and precipitation classification, (c) defining an alternative method to monitor dynamics of snowpack with high temporal resolution. The paper will be organized in the following sections: case studies description, method and

algorithm definitions, results and discussion, conclusions.



## 2. CASE STUDIES

### 2.1 SODANKYLÄ INFRASTRUCTURE

#### 2.1.1    Test Site

As the first case study, we focused on a test site in Sodankylä infrastructure which is an essential part of the Arctic Space Centre of the Finnish Meteorological Institute (FMI) (Figure S1).

In particular we referred to the images from the MONIMET Camera Network (Arslan et al. 2017; Peltoniemi et al. 2018, http://monimet.fmi.fi). It is a network of digital surveillance cameras mounted on 14 sites, each site having 1-3 cameras. Concerning on our purpose, operators of Finnish Meteorological Institute, systematically collected snow depth (SD) time series

and snow water equivalent (SWE) since 1909 (Leppänen et al. 2016). Moreover in 2006 the manual snow survey program expanded to cover snow macro and microstructure from regular snow pits at several sites using both traditional and novel measurement techniques.

In Sodankylä there are a lot of cameras, but we selected the Peatland one because: it was not affected by snow canopy interception, it is located in the prairie, it does not suffer from very strong windy conditions, so the relative position between

camera and stake remains the same (because it is fundamental to keeping stake in the same position for the whole retrieving period to correctly estimate the snow depth). In addition, at the same site there is also an automatic weather station as reported in Figure S1.The peatland field was established in 2003 for UV measurements that lasted until 2009. Manual snow pit measurements were made at the peatland during the period from 2009 to 2015.

#### 2.1.2    Data Description

The implemented algorithm can estimate snow depth with a counting pixel procedure based on image reading. Snow depth temporal resolution depends on the scan rate of the camera and much more strongly to the image clarity. Moreover, not all the images can be useful to estimate snow depth. In fact, some images were collected when the sun had already gone down, the light was very poor, and images appear totally black. Images were saved at 30 minutes resolution from 8:00 AM to 6:00 PM,

but generally in winter the first three images and those referred to the hours after 15:30 are totally black, so in this case the algorithm cannot capture the snow depth. For this reason, we decided to limit the use of images within a period of about 6 hours with a time resolution of 30 minutes.

Although the camera at peatland field was positioned in 2014, the first available image to detect snow depth with the help of a snow stake was related to the 6th November 2015. The last one was at the end of April 2019, when the snowpack was totally

disappeared.

The camera was located in front of a white snow stake with $1 \cdot 10^{-2} m$ spacing black markers. In this case on the spacing it depends on the resolution of the snow depth estimation because the algorithm cannot detect anything less than $1 \cdot 10^{-2} m$. Another important factor is the pixel resolution, which corresponds in this case to 2592 x1944 pixels with horizontal and vertical resolution of 96 dpi. We highlight also that images from 2015-2016 can be used to estimate snow depth with two or

three different stakes, positioned at different distances from the camera. The positioning of more than one stake enables to study the snow depth spatial distribution and also to compare the estimations between different stakes. In fact, the proposed method can fail, principally due to some obstructions which limited camera's view, or in case that shadows appeared on the ground in front of the stake, it can be picked as a marker. So, in this case we can find a strong underestimation of the snow depth, but this underestimation could affect only one stake and not the others, so, comparing different stakes we can remove

errors and misclassifications.

The ultrasonic snow depth sensor, available in the test site, which collected data every minute was used to validate and compare the results of the estimations of the proposed method. Generally, ultrasonic measurements suffer from a lot of noise due to the





conversion of the travel time of the emitted and retrieved signals or the grass above the ground. A correction was automatically implemented and was based on air temperature data. To compare the results and reduce the noise we defined a snow depth

value every 30 minutes. This dataset is available from the Finnish Meteorological Institute Arctic Space Center Website (http://litdb.fmi.fi/suo0003_data.php), from 15th October 2010 to 27th March 2019.

Moreover, operators and researchers of FMI planned weekly field campaigns with the aim of retrieve snowpack conditions from 9th October 2010 to 28th April 2015. They measured snow water equivalent, density and snow depth. In Figure 1 we plotted the measurement, together from Sodankylä peatland field, with black dots, representing the ultrasonic snow depth

sensor and red dots, representing the manual measurements. Having a fine temporal resolution, no lack of data for a long period, these snow depth measurements could be considered as a valid reference to test our method.

### 2.1.3    In-situ measurements comparison (Optical Sensor Validation)

In the period between 2009 and 2015, FMI had collected manual measurements of the snow depth, using rulers, with field campaign planned every week. As reported in literature (Pirazzini et al. 2018) manual measurement with rulers is the historical

method to retrieve the snow depth on the ground. We compared the ultrasonic snow depth measurements with the manual ones. We wanted to see how the ultrasonic measurements performs compared to the manual ones. This is important because the manual measurements are available until 2015 but the ultrasonic measurements are available until 2019.

This validation procedure gives a possibility to compare estimations of the proposed method with the ultrasonic measurements between 2015 and 2019. The Ultrasonic sensor collected snow depth with 1-minute resolution. In order to reduce noises and

define a time series comparable with the time-lapse photography method we aggregated the ultrasonic measurements at hourly resolution. In Figure 1 we presented the comparison results between manual measurements (red dots) and ultrasonic one (dark dots). Focusing on the years 2011 and 2014 we found high agreement between the two methods in terms of trend and peaks, whereas in 2012 and 2013, we can see some differences. In 2012, ultrasonic snow depth sensor was characterized by some problems, prevented the measurements from being performed correctly.

In 2015, ultrasonic snow depth sensor measured the highest snow depth peak which was smaller than the manual measurements. This situation was vice versa in 2013. In both 2013 and 2015, the accumulation season (typically winter), and the melting one (typically late spring) show a good agreement although there were differences in estimation peak values. In addition, we highlight that snow depth ultrasonic sensor were not affected by a persistent under/over estimation that can be a warning of instrument drift. Finally, for each day in which we had both manual and ultrasonic snow depth sensor

measurements, we calculated RMSE (Root mean square error, as defined in the next section) which was equal to $4.531 \cdot 10^{-2} m$. By this validation procedure we retained reliable ultrasonic measurements and we used these measurements as reference for the years between 2015 and 2019 to validate the estimations made through the FMIPROT. Table 1 reports information about sensors, period of time and time resolution used.

### 2.2  GRESSONEY DEJOLA (ITALIAN ALPS)

### 2.2.1 Test Site

The first Italian case study is located in Valle d'Aosta region, at an altitude of 1850 m (a.s.l.) in which from 1927 the Meteorological Italian Society has measured snow depth on the ground by rulers. This site can be considered a historical meteorological observatory of the snow depth retrieval in Italy due to the long data availability.

In Figure S2, we reported (from the right to left), Valle d'Aosta inside the Italian territory, Gressoney la Trinitè Dejola site,

and a picture of the snow stake inside the meteorological observatory.

Recently, from 01/09/2013 AINEVA (Interregional Association for snow and avalanche problems coordination) has positioned a snow stake with 0.01 m spacing dark marker. The total length of the stake is 2 m and has a width of $10 \cdot 10^{-2} m$. As underlined by Figure S2, the snow stake was not positioned specifically for the remote sensing image processing purpose and



it is affected by a not optimal geometrical configuration because there is a different distance between the camera and the top
and the bottom of the stake. Importance of the positioning and optimal geometrical configuration will be described later in the
section related to the discussion of results.

### 2.2.2 Data Description

AINEVA, from 01/09/2013 saves images from the camera at hourly resolution from 6:00 AM to 6:00 PM. Especially in the
winter season some images appear totally black, so we decided to use only those taken from 8:00 AM to 15:00 PM.
AINEVA operators made the visual estimation of the snow depth, saving in an excel file the snow depth value accumulated
on the ground. From a personal communication with people who made these measurements, the camera is affected by a parallax
effect, due to the relative distance between the height of the camera and the bottom of the stake In addition, the real snow depth
value cannot be retrieved simply reading the first marker not covered by snow because there is a heat transfer from the stake
to the snowpack due to conduction and reflection of incident solar radiation. So, measurements reported by operators
considered this effect, defining the real snow depth obtained as a cross between snow stake markers and a virtual line which
links the snowpack accumulated on the ground at the left and right side of the stake (Figure 11a).
In this way AINEVA operators defined two snow depth values for each day at 8:00 AM and 2:00 PM. We decided to use these
values as a reference to compare the results obtained by our method. In Figure 2 we plotted for each day these two estimated
values from 01/09/2013 to 20/05/2019.
As we can see for the years 2013-2014 and 2017-2018, snow depth reached a value of 2.5 m overcoming the maximum length
of the stake. In addition, many images are obstructed by snowflakes which remained attached on the camera's view. So, in this
case, they used other information to define the snow depth time series above the ground. Because our aim is to compare and
validate the snow depth estimated using digital images, we decided to discard years in which the reference value cannot be
detected directly by visual inspection of images, focusing only in the period between October 2014 and December 2017.
Dimension of the images are 1280x960 pixels, with horizontal and vertical resolution of 96 dpi. Table 1 reports information
about sensors, period of time and time resolution used.

## 2.3  CARESER DAM (TRENTINO REGION – ITALIAN ALPS)

### 2.3.1  Test Site

The second Italian case study is located in Trentino region, the Careser dam, at an altitude of about 2600 m, near Careser
glacier. The Civil Protection Agency positioned a snow camera in 2014 with the aim of collecting snow data for hydrological
purposes. In addition, AINEVA (Interregional Association with the aim of coordinating and documenting the problems related
to snow and avalanche) and Meteotrentino collected and spread daily information related to the snowpack such as snow depth
and density to define the avalanche risk with publishing daily bulletin available on the website to inform people. These sites
are well widespread, covering the whole region both in space and altitude.
In Figure S3 we reported the location of the Careser dam and an image from the camera.

### 2.3.2  Data Description

Despite the fact that the camera was not specifically positioned for a remote sensing image processing on estimation of snow
depth, this can be considered as a valid example to test our method. The camera was placed very near to the stake with width
of $15 \cdot 10^{-2} m$, and colored in yellow with $1 \cdot 10^{-2} m$ spacing markers. The total length of the stake is 2 m.  Unfortunately,
the temporal resolution is poor in this case, as they collected only 4 images per day at 5:00, 8:00, 11:00 AM and 2:00 PM. In
most of the cases the first one at 5:00 am appears totally black, so we have discarded it.



The observation period goes from 01/01/2014 to 31/12/2018, but in some periods, there was a lack of data due to bad atmospheric conditions in which strong wind produced a camera rotation resulting images without snow stake. Other images

which had to be discarded are those in which snowflakes remained on the camera's objective, making images totally grey or snow stake partially invisible. These problems occur when the camera was not protected properly. In fact, in most of the places where this technology was implemented, the camera was placed into a small wooden box. In addition, sometimes, the snow stake is positioned near a fence, that, especially in the melting season, enables a percolation toward the stake, which increases the snow melting.

After the summer 2017, AINEVA replaced the camera with increased pixel resolution one. As a result of this change the relative position between camera and stake was not the same for the whole observation period.

About images, from 2014 to the middle of 2017, they have 768x576 pixels with horizontal and vertical resolution of 96 dpi. Starting from November 2017, the number of pixels of each image was increased to 1920 x1080.

Unfortunately, in this case we have not any snow depth sensor or snow field manual measurements, so to have a reference

value to compare our estimations, we made visual estimations of the snow depth, simply opening each image, and watching the snow stake markers.

In many cases the visual estimations were not so easy because an amount of snow remains attached on the snow stake; another problem is the differential melting rate between snowpack close to the stake and at a distance of more than 1 m. This is caused by the high snow stake reflectivity related to the incoming sunlight, heat transmitted by conduction and the water percolation

from the top to the bottom of the stake. This effect is hardly quantifiable and depending on the atmospheric conditions. So also, for the "visual estimation", this time series is affected by a noise which can be quantified in $5 \cdot 10^{-2}m$.

The "visual estimation" is obviously time consuming, because we have to open and read each image, but we are pretty sure that this kind of estimation is not affected by outliers or unreliable values. We have also to point out that the "visual estimation" is subjective and the same image can lead to different values by different observers. In Figure 3 we reported the snow depth

visual estimation for the period between 1st November 2015 and 31st December 2018. Table 1 reports information about sensors, period of time and time resolution used.



## 3. METHOD

In this section it is described how camera images (which show a snow stake with graduated markers) can be used to estimate the snow depth subdivided in the following points:

    I.    FMIPROT: images processing tool developed by FMI;

    II.    Snow depth retrieval procedure: procedure developed in FMIPROT to retrieve snow depth estimates;

    III.    Snow depth estimation algorithm: it will include all the processes and calculations performed on a single image to
calculate the snow depth, if a snow stake with graduated markers is visible. In addition, we will describe which are the parameters that must be defined to run the simulations;

    IV.    how to build a reliable time series without NaN (Not-a-Number) and biases;

    V.    efficiency and errors of the procedure: how accurately the snow depth was estimated.

### 3.1 FMIPROT (Finnish Meteorological Institute Image Processing Tool)

In order to estimate the snow depth from digital images in which are visible one, or more than one snow stakes with $0.01\ m$ spacing markers, we used the snow depth retrieval algorithm implemented into the FMIPROT which calculates snow depth based on a pixel counting algorithm routine.

FMIPROT (Tanis et al. 2018) was developed for analyzing digital images from multiple camera networks for various applications such as vegetation phenology and monitoring of snow cover. This toolbox has a user-friendly graphical user
interface (GUI) and can be used easily without any specific knowledge. Interface allows to easily download images from cameras network (online or offline) which provide continuous and widely series with the aim to monitor some environmental features such as phenology and snow.

Current features are automatic image downloading and handling, GUI based selection a region of interest (ROI), automatic analysis chain, GUI based plotting, creation of HTML reports with results on interactive plots, ROI based indices such as green
fraction index (GF), red fraction index (RF), blue fraction index (BF), green-red vegetation index (GRVI), green excess index (GEI), snow cover fraction estimation with geo-rectification, snow depth estimation and time-lapse animation. FMIPROT is freely available at the website http://fmiprot.fmi.fi.

In this work we used a protocol already implemented by FMIPROT, which allows to easily process images; with this tool we can select the so called "Snow Depth algorithm", developed by FMI researchers and presented for the first time by Tanis et al.
(2018) to calculate snow depth.

### 3.2 SNOW DEPTH RETRIEVAL PROCEDURE

Inside the FMIPROT to estimate snow depth, we followed procedure given below (Figure 4):

- Add a camera network from a single wizard: all the images for the same case study must be collected in a single folder. In each folder each image is called with the same code, a part of that must be referred at the data acquisition
time. For example, for Sodankylä Peatland case study, one image it's called: "*sod_pin_peatland_20170101_090135*" where the number means respectively: year, month, day, hour, minute and second of the time acquisition.

- Region of Interest (ROI) definition: After the camera selection, we must define the region of interest, drawing a polygon on a reference image (without snow cover) which start from the bottom to the top of the snow stake or a given level with known depth. This polygon will be used to cut each image focusing the calculation only inside of
that;

- Select the algorithm: under the section "analysis" within FMIPROT, depending on which feature we are interested we can select different algorithms: Color Fraction Extraction, Vegetation Indices, Georectification etc.; here we selected the snow depth algorithm which, originally, was named:"SNOWDEP001", which allows to estimate snow depth if images show a graduated snow stake;





●   Define the analysis parameters: each algorithm has its own parameters. In this case we must define:

       o   $L_S$: the total length of the selected ROI, with the corresponding measurement unit (m);

       o   $T_S$: the brightness temperature threshold: which defines a pixel as black or white comparing its value with the same threshold;

       o   Gaussian filter σ, which specifies the width of an area near each pixel used to detect the marker position;

375        o   The bias factor B, if the selected ROI includes also grass, we have to consider that snow depth has not to be referred to the 0 value, but we would have a bias of $0.05\ to\ 0.1\ m$, depending on the grass height. Generally, it is better to define the ROI starting from the first visible marker above the grass level;

   ●   Define the filtering parameters: To reduce the computational time required to make the calculations, it is better to define some parameters which allow to discard some images which can be considered not useful. For example, images

which appeared totally black did not give any information, so they were removed defining a minimum brightness of 30%. At the same time, images in which we have a strong sunlight influence, the last one can reduce     the brightness contrast between markers and snowpack which bring an over/underestimation of the snow depth. Inside FMIPROT GUI, there are three different groups of thresholds: Image threshold, ROI Threshold and Pixel Threshold. About the whole image we can define a minimum and maximum level of brightness (B) on a scale of 0 to 1 normalized by the

mean brightness value of each image. The second parameter is the luminance (LU), also in this case we can define a range of possible luminance values. This parameter is calculated as a weighted mean of Red (RC), Green (GC) and Blue Channel (BC) as: $LU = 0.2989 \cdot RC + 0.587 \cdot GC + 0.114 \cdot BC$

      Within the ROI (Region Of Interest) we can define a possible range for each color fraction: for example, for the blue channel its fraction can be defined as: $BF = BC/(BC + RC + GC)$.

The last parameter is the Pixel Threshold (PT), with which we can discard pixel characterized by particularly low or high values of Red (PTRC), Blue (PTBC) and Green channel (PTGC).

     For our computations, generally we defined these threshold parameters, which allow to preprocessing images, discarding those which were not consistent with these parameter ranges: $0.3 < B < 0.7$ , $0.3 < LU < 0.7$, $50 < PTBC < 100$.

●   Selecting images by date and time of day: We can run the analysis for all the images within the folder or we can specify a time period range and/or a specific hour for each day.

     First of all, FMIPROT checks whether images can be used for the estimation of snow depth or not, with the parameters within predefined threshold range. Otherwise, it will discard those. Secondly, it runs the algorithm (snow depth algorithm in this case) for the single image separately estimating the snow depth. Finally, the results were reported in a txt files in csv format,

subdivided in two columns: the first one is the date, and the second one is the snow depth values estimated.

     Here we have mentioned only those parameters or threshold which we have used in our computations, for additional information or more details about FMIPROT can be found in http://fmiprot.fmi.fi.

     In the following section we will describe the snow depth estimation algorithm, written in python language.

**3.3 SNOW DEPTH ESTIMATION ALGORITHM**

Inside FMIPROT there are many algorithms already implemented to retrieve some environmental features, such as the fractional snow cover, the vegetation index, the geo-rectification. In 2018, Tanis and Arslan wrote an algorithm using some predefined images processing function in python language to estimate snow depth based on images with a snow stake with some markers. This algorithm can be defined as a remote sensing automatic algorithm which substituted the traditional reading of the snow stake on-site. The proposed algorithm used image segmentation to detect where snow surface intersects the snow

stakes. This algorithm was tested using images from spice site (within Arctic Space Center) in Sodankyla, and more details





about the testing phase can be found Tanis et al. (2018). The snow depth estimation algorithm (clearly explained in Figure 5) is subdivided in these following steps:

1) ROI IDENTIFICATION: starting from an image without snow cover, with FMIPROT we selected the region of interest drawing a polygon, which defined a contour around the snow stake. So, the pixel counting algorithm clips each image considering only pixels inside of this. In addition, we must define the real length of the whole ROI in m (for example, $L_S = 2.00\ m$ for the Careser dam case study). In most of the cases, it is better to define a polygon which includes the whole stake from the ground to the top level, but it is not mandatory, we have just to guarantee that the maximum snow depth value is less than the top of the ROI.


    2) GAUSSIAN FILTER APPLICATION: at each image it can be applied a Gaussian Filter based on the parameter σ, which smooths the brightness difference, reducing pixel noises.

    3) THRESHOLDING: comparing the pixel value with a predefined brightness temperature threshold $T_S$, we can classify
425       each pixel as black or white (0 or 1). This threshold can vary widely, selecting different observation period, and we must define the best one able to reduce misclassifications. Related to this parameter is better to have a strong reflectance difference between the stake background and markers. We found that the best coloration of the stake is yellow or white for the background and dark color (i.e. black) for the markers.

4) SEGMENTATION: it is the process of partitioning a digital image into multiple segments. In our case this process defines the marker's contour with the edge detection. That's why is fundamental that stake must be graduated. In fact, the snow depth estimation resolution depends on the marker spacing. Markers with little distance can improve estimation accuracy, but we suggest remaining above 0.01 m because they must be identified clearly considering the distance between stake and camera.


    5) SHAPE FILTERING: it consists on characterizing and filtering objects in binary scenes by its shape. In this case shape filtering allows to calculate the pixel row for each marker ($d_i$). Note that the counting started from the upper part of the image. 0 means the top of the ROI.

6) CALCULATING EACH MARKER HEIGHT: knowing the marker row, we can calculate the number of pixels from the ground $H_i$, simply subtracting the whole stake length $H_S$ (in pixel) with the above calculated marker's row $d_i$.

    7) CALCULATING THE POSITION OF THE FIRST VISIBLE MARKER: the snow depth layer is located near the first visible marker from the bottom of the stake, which is characterized by the minimum height value. Its pixel row
445       is defined as:

$$S_D = min(H_i)\ \ [N°\ of\ Pixels] \tag{1}$$

    8) CONVERTING THE PIXEL DISTANCE IN SNOW DEPTH REAL VALUE: the number of the pixels must be converted into the real depth multiplying it with the ratio between the real snow stake length ($L_S$) in meter and number of the pixel of the whole stake ($H_S$):

$$SD = S_D \frac{L_S}{H_S}\ [m] \tag{2}$$





### 3.3.1 SIMULATION PARAMETERS

Counting pixel algorithm routine required the parameters definition, in particular for each case study we have to choose the best threshold ($T_S$) and the gaussian filter (σ), which allow to better estimate snow depth; the best parameters configuration is that allows to increase the brightness contrast between snowpack layer and stake's markers. Threshold can vary widely, because images have different resolutions, due to the different geometrical configuration and especially related to the different visibility conditions in terms of brightness and luminance: Sodankylä is characterized by low visibility levels, Careser dam, especially in the hours near 12, sometimes it is affected by high reflectance of the sunlight, much more stable in terms of visibility level can be considered the Gressoney Trinitè Dejola site because it is located inside the forest and without any direct solar incoming radiation.

The same thing happened for the gaussian filter σ, which can assume values from 1 to 10. Firstly, we run the simulation in a small period (10 days) searching for the best parameter configuration able to reproduce the snow depth visual estimated. After having found these values we run the simulation for the whole observation period.

In Table 2 we reported the defined parameters used in our computations. As we can see, depending on the maximum snow depth observed on the ground, we defined different ROI length: Careser dam is located near 2600 m a.s.l. so the expected maximum snow depth is near $2\,m$, at contrary at Sodankylä Peatland site, we observed a small value equal to $1\,m$. Related to the brightness temperature we used values from 65-70, and for the gaussian filter from 1 to 7.

As we will show in the next paragraph, the best method consists of repeating simulations with different parameters combination and defining an "ensemble" snow depth as the mean of those. We will show how this technique can improve results, obtaining a more reliable snow depth time series using Sodankylä Peatland images at hourly resolution, focusing on a period of 6 months. This can be generalized to all case studies, but here we reported results only for one, due to the high computational time required to repeat simulations.

### 3.4 BUILDING A SNOW DEPTH TIME SERIES WITHOUT NOT-A-NUMBER VALUES

When we have high temporal resolution (1 hour), and a long period without lack of data, as in Sodankylä case study, we can define a procedure to build the whole snow depth time series without not-a-number or biased values. This is very important for hydrological purpose, but until now this algorithm was tested only for small periods (months). If we think about the possible implications of snow depth measurements, such as, input to a hydrological model, to define the hydrological balance of a watershed, is better to have the whole snow depth time series as long as possible. Moreover, removing "0" and Not-a-Number (NaN) values, allows to define also snowfall events with high accuracy, simply differentiating snow depth time series.

So here the principal aim is not only the punctual definition of the snow depth but also defining a snow depth time series with high accuracy. We highlight that this procedure will be defined without any snow depth measurements knowledge, in other terms, without data assimilation, because in most cases we have only images without real measurements.

FMIPROT algorithm provides snow depth estimations near the real values with high percentage, but sometimes can fail, especially when atmospheric conditions were bad. In these cases, we found snow depth estimated values near 0 or classified as Not-a-Number or affected by high biases. So, we defined a procedure to remove these incorrect values, running simulations five times with different parameters' combinations. In particular, we fixed the threshold brightness temperature parameter at 70, and we defined 5 different values of the gaussian filter parameter σ from 1 to 5. Then each time series was used to define the "corrected snow depth time series". In particular we define a procedure that can be subdivided in these following points:

1) **Loading the single snow depth time series** estimated with fixed threshold Ts and gaussian filter parameter σ, called as: $SD_O(t)$, where "O" means original.

2) **Removing errors and high biases** obtaining a more accuracy snow depth time series, that we will call as correct: $SD_C(t)$.



First of all, we suppose that in one hour was difficult to have a snowfall or melting much more than $0.02\ m$, so we reclassified as Not-a-Number $SD_O(t)$ if we have a difference more than $0.02\ m$ to the previous or the following one. In some cases, we found a cluster of misclassifications, so values near Not-a-Number were suspicious, so we decided to reclassify each value near Not-a-Number as Not-a-Number itself. Summarizing we can write these simple conditions:

$$if\ (|SD_O(t) - SD_0(t-1)| > 2\ cm\ or |SD_O(t+1) - SD_0(t)| > 2\ cm) \rightarrow SD_C(t) = NaN \tag{3}$$

$$if\ (SD_0(t) = NaN\ ) \rightarrow SD_c(t-1) = Nan\ \&\ SD_C(t+1) = NaN \tag{4}$$

In addition, as commonly used when we worked with optical snow depth sensor, to remove noises, we correct value at time t if this value had an absolute difference more than $0.005\ m$ (S) from the mean of the 12 (WL) following and previous values, so we substituted the value at time t with a moving average of 24 values centered at time t. If we have NaN values inside this window, we did not consider them:

$$if\ \left(\left|SD_C(t) - \frac{1}{n}\sum_{t-WL}^{t} SD_C(t)\right| > S\ \&\ \left|SD_C(t) - \frac{1}{n}\sum_{t}^{t+WL} SD_C(t)\right| > S\right) \rightarrow SD_C(t) = \frac{1}{n}\sum_{t-WL}^{t+WL} SD_C(t) \tag{5}$$

Where S is the maximum snow depth difference, equal to $0.005\ m$ and WL is the window length of the moving average procedure, equal to 12 values. In this case n can be different from WL, depending on how many NaN we found between t-WL and t+WL. In Figure 6 we reported results of these two previous steps, using images from Sodankylä Peatland with hourly resolution, from 1st January to 30th June 2017, and FMIPROT estimations obtained with parameters: $Ts = 70$ and $\sigma = 5$.

3) **Defining the final snow depth time series**: The same correction (points 1 and 2) was repeated for each snow depth time series with 5 different values of parameter σ, from 1 to 5, which can be called as $SD_{C\ i}(t)$, where c means corrected and i goes from 1 to 5. The advantage of having 5 simulations, was the possibility of compare the single value of one simulation $SD_{C\ i}(t)$ with the mean of the others 4, classifying as NaN the first if the difference was more than $0.001\ m$. In this way one single value was defined as reliable only if in the other 4 cases we observed more or less the same value. Summarizing we wrote this condition:

$$if\ \left(\left|SD_{C\ i}(t) - \frac{1}{m}\sum_{j \neq i} SD_{C\ j}(t)\right| > 0.1\ cm)\right) \rightarrow SD_{C\ i}(t) = NaN \tag{6}$$

Where m is equal to 4, i can take a value from 1 to 5, and j must be different from i. In this case m can assume values from 1 to 5 depending on how many values are different from NaN or 0.

Then the "Mean Snow depth time series" was obtained as the mean value of the others 5:

$$SD_{MEAN}(t) = \frac{1}{n}\sum_{i=1}^{5} SD_{C\ i}(t) \tag{7}$$

This time series still contained Not-a-Number Values, so, thanks to the temporal resolution of 1 hours and not having a long time period with lack of data, we reclassified Not-a-Number values as the first previous valid values:

$$if\ SD_{MEAN}(t) = NaN \rightarrow SD_{MEAN}(t) = SD_{MEAN}(t-1) \tag{8}$$

We underline that this condition works well only if images have fine temporal resolution and small period with lack of data. In Figure S4 we reported with colored dots the 5 corrected time series obtained with a fixed brightness temperature threshold $T_S$ and for different parameter σ from 1 to 5, after removing bias procedure previously defined (points 1 and 2), whereas with black dots and line the $SD_{MEAN}$ time series as a result of the points 7 and 8. The simulation period was the first half of 2017





and images are those collected from Sodankylä Peatland camera. Focusing on the $SD_{MEAN}$ time series we showed that, despite
that the original simulations were characterized by lack of data or NaN values we defined a complete hourly time series. This
procedure can give better results if we had images with high temporal resolution (less than one hour), because we can't observe
high drop between two subsequent values also in case of high melting or snowfall rates, making the discarding phase much
easier. Within the results section, this procedure will be evaluated in terms of accuracy and efficiency.

### 3.5 ACCURACY PARAMETERS

To better understand if this method is comparable in terms of accuracy and efficiency to the most common ones available in
the literature, we calculated two indexes: Root Mean Square Error (RMSE) and Nash Sutcliffe Efficiency (NSE). Those
indexes compared simulations results obtained by FMIPROT, which will be called with the subscription "Sim", with the
Observed snow depth values (Obs). The last one was obtained by images inspection for all the three case studies. In the future
we plan to compare results also with in-situ measurements. We remind the definition of the two above mentioned accuracy
parameters, even though are commonly used inside the hydrological community:

$$RMSE = \sqrt{\frac{1}{n} \sum_{i=1}^{n} (SD_{Sim}(i) - SD_{Obs}(i))^2} \qquad (9)$$

$$NSE = 1 - \frac{\sum_{i=1}^{n} (SD_{Sim}(i) - SD_{Obs}(i))^2}{\sum_{i=1}^{n} (SD_{Obs}(i) - \overline{SD_{Obs}})^2} \qquad (10)$$

where i means the temporal resolution, n is the total number of simulated values, $SD_{Sim}$ is the simulated Snow Depth, and
$SD_{Obs}$ the Observed one (obtained by Visual inspection of images). Moreover, $SD_{Obs}$ is the mean observed value.

RMSE allows to calculate how much the single snow depth estimated value was different from the observed one, in average
terms inside the whole observation period. It will be used to check the result's accuracy.

NSE compared the residual variance (numerator) with the observation variance (denominator). An efficiency value of 1
corresponds to a perfect match, but also values from 0 to 1 indicates that simulations are better predictors than the mean
observed value. Generally, to indicate a sufficient quality has been suggested values from 0.5 to 0.65.

Unfortunately, this coefficient is sensitive to extreme values, so it is not optimal in case of observed values with high biases.





## 4. RESULTS

Here we reported the results obtained by FMIPROT algorithm previously defined. Unfortunately, even if images appear clear, the algorithm can fail in the snow depth detection since the geometrical configuration between camera and stake in all case study was not predefined of this aim. In fact, especially for the two Italian case study, these images were used only as a reference but are not widely exploited for monitor the snowpack dynamic.

Related to the snow depth reference values, used to compare the FMIPROT estimations, for Sodankylä Peatland case study in addition to the visual estimations from images, we have also ultrasonic sensor and manual measurements. For the two Italian case studies, we must refer only on the visual estimations from images. In particular for Careser dam, we haven't any additional information, but for the Gressoney Dejola, the Meteorological Italian Society, visually estimated snow depth, with a "critical" snow depth definition, which means that they took in account the high melting that occurred close to the stake, and tried to remove the parallax effect, due to the different height between camera and the top of the stake.

### 4.1 SODANKYLÄ PEATLAND

Here we will discuss results about Sodankylä Peatland case study. Probably, compared to the two Italian case study, in this case the algorithm works well because the stake was positioned near the camera, the images had an enough pixel's resolution and appeared clear, without high or low values of brightness and reflectivity. Moreover, it was positioned inside a small building which can protect it against strongly windy conditions and possibility of obstruction of the camera's view caused by snowflakes.

If we have to suggest the proper geometrical configuration between camera and stake defining a " design " snow field, we highlight the importance of positioning more than one stake, all at the same distance from the camera, and choosing the best observation angle which allows to see each stake marker with the same pixel subdivision, reducing the parallax effect.

In this case, the stake much closer to the camera was enough to retrieve with enough temporal resolution and geometrical accuracy snow depth values. As mentioned in data description section, here we had images only after 2014, so to check the reliability of the FMIPROT snow depth estimation, we can only use the ultrasonic snow depth sensor, positioned near the camera, because we didn't have any other information, like manual measurements. The ultrasonic snow depth reliability was showed in the data description section, so we can consider these measurements as reliable.

The comparison between simulations and observations of the snow depth is reported in Figure 7, related to the period from 6/11/2015 to 27/03/2019. In particular we plotted: with blue dots FMIPROT simulations, with red dots visual estimation of images, with black dots Campbell ultrasonic measurements. In this case, the snow depth time series is correctly reproduced and good agreement between simulations and measurements has been found.

Generally, visual estimations provide greater values, so we can observe an underestimation of the ultrasonic snow depth sensor and FMIPROT estimations. Here it is interesting to highlight that the temporal resolution of FMIPROT estimation is hourly, within the daylight which starts from 9:00 AM and ends at 3:00 PM. Ultrasonic sensors measured snow depth at 10 minute resolutions, and visual estimations were reported weekly. So, if the images were clear and without a long period with lack of data, we can build snow depth time series with high temporal resolution. About the accuracy, we calculated RMSE and NSE, comparing FMIPROT estimations, with Ultrasonic Snow depth measurements and visual observation of images.

In this case, the simulations and ultrasonic snow depth sensor measurements were compared, and we obtained $RMSE_{FMI-US} = 9.6151 \cdot 10^{-2}m$, $NSE_{FMI-US} = 0.7991$. However, comparing FMIPROT estimations with visual observation of images, $RMSE_{FMI-VO} = 8.4758 \cdot 10^{-2}m$, $NSE_{FMI-VO} = 0.8814$. Visual observations are considered much more reliable because were obtained by a direct reading of the markers stake instead of the ultrasonic measurements which is positioned much nearer to the camera and very far from the stake. We have also to highlight that the comparison with Ultrasonic Snow Depth Sensors was done at hourly resolution, despite that those with visual inspections have weekly resolution. Another interesting factor is



the number of misclassifications, or the inability of the FMIPROT algorithm to give a snow depth real value. In this case, NaN values were 2.91% of the images.

**Building the snow depth time series in the first half of 2017**

Thanks to the hourly image resolution and a period without lack of data, we tested the "smoothing" procedure defined in section 3.4 inside a 6-month period, on the first part of the year 2017, to obtain a complete snow depth time series without NaN values. Moreover, we repeated simulations fixing the threshold parameter at 70, and varying the gaussian filter parameter σ from 1 to 5. Each simulation was cleaned and then compared each other verifying the coherence from each of them. The

final snow depth time series was obtained simply doing the mean among these five simulations. In Figure 8, we reported the comparison between the FMIPROT snow depth time series, visual estimation from images and the ultrasonic snow depth measurement by the Campbell Instrument.

As we can see from the picture, we can build the snow depth time series with high temporal resolution and with high accuracy. This fact is shown if we calculate the accuracy parameters also compared with those obtained by the ultrasonic snow depth

measurements. As previously, we considered as reference the visual snow depth estimations. Related to the ultrasonic snow depth sensor we have: $RMSE_{US} = 5.1706\ cm, NSE_{US} = 0.881$. But for the FMIPROT Estimated Snow Depth we have: $RMSE_{FMI} = 3.951\ cm, NSE_{FMI} = 0.9166$. With these values we can say that FMIPROT is much more accurate and efficient compared to the ultrasonic snow depth sensor, and especially this happened in January and April.

Those results are very good because we can define not only the punctual value of the snow depth with high accuracy, but we

can build the snow depth time series without lack of data. Moreover, those results were obtained without data assimilation procedure, which in most of the cases "force" the results knowing the real snow depth value. In addition, this procedure is completely automatic and low time consuming. This complete time series also, can be very useful for hydrological model to better define the snow depth on the ground that constitutes, especially in mountain regions, a large part of the hydrological balance.

The last implication could be that of using these simulations to retrieve snowfall events in dangerous avalanche regions, where in-situ measurements are dangerous and or in remote areas difficult to reach (very steep terrain, mountain glacier, ice sheet).

**4.2 GRESSONEY LA TRINITE'**

Here we reported the results for Gressoney La Trinitè Dejola case study. Inside the observation period which spans from 10/12/2013 to 20/05/2019 we selected the period from: 04/11/2014 to 31/12/2017 because in the other years the snow depth

observations from AINEVA operators (Figure 2) show values much more than 2 m (stake length). In these cases, algorithm cannot retrieve correctly the snow depth, due to the fact that the snowpack overtakes the ROI, so we decided to discard a-priori them.

About the snow depth observed values, which will be used to compare our results, AINEVA operators for each day defined three values with a visual inspection of the images, respectively at 8:00 AM, 2:00 PM and 9:00 PM. The last one is equal to

the last visible image for each day (AINEVA operator personal communication) so, did not know exactly at which time those measurements were referred, we avoided using those. For these reasons, having only 2 reference values for each day, to estimate the snow depth with FMIPROT we decided to select only two images per day, which corresponds at the same hours (8:00 AM and 2:00 PM). In most cases images appeared clear, without any lack of data, and defining the threshold values as mentioned in the algorithm section, we made the computations, using FMIPROT.

In addition, as mentioned in section 2, AINEVA operators defined a visual observation that considered the parallax effect, so, the algorithm detection of the first visible marker could be different from this visual estimation as shown in Figure 11a.

In this case, we had to define two different parameterizations to make simulations due to the parallax effect, caused by the different distance which exists between the top and the bottom of the stake to the camera. In fact, here the relative position between camera and stake was not optimal, because the camera was not positioned at the middle of the stake in terms of height.



630 For this reason, if we considered the same real stake length, such as 0.1 m, at the top of the stake was represented inside a greater number of pixels, compared with the bottom. As the consequence if we considered a unique ROI the parallax effect caused a strong underestimation. In conclusion, when the snow depth was less than 0.3 $m$ we used this parametrization: $L_S = 0.3\ m$ , $T_S = 70$, $\sigma = 4$.

When the snow depth was greater than 0.3 $m$ we have to define a ROI (region of interest) which contained the maximum

635 observed snow depth: $L_S = 1.2\ m$, $T_S = 70$, $\sigma = 4$. Summarizing, between the two computations, the difference was only in the Region of Interest definition.

Comparing the simulations obtained by FMIPROT and the interpreted ones given by operators, we found that there was a strong difference between the two as reported in the scatterplot in Figure 9. The scatterplot showed a constant underestimation of the FMIPROT algorithm results, for this reason, in the following we will define a constant bias to correct our estimations.

640 In addition, we calculated the ratio between observed and estimated values, as reported in Figure 10 in which we can see that the lowest values are those affected by the greater underestimations: this fact was expected because, due to the high melting near the stake, if there was snowpack accumulated on the ground, the first visible marker could be near 0. In other terms when the real snow depth on the ground is near $0.1m$ algorithm can fail. To better explain this fact, we reported also an image related to the data: 13/04/2016 (Figure 11b).

645 Here the estimated snow depth reported by operator was: 0.24 $m$, but the algorithm gives 0.0516 $m$ because the stake was almost totally clean. As we will discuss in the following section, when the snow depth is less than 0.1 $m$, those images, and this geometric configuration didn't allow to estimate adequately snow depth. But we have to highlight that this is due to the imperfectly design of the relative position between camera and stake and not to the algorithm incapability.

So, these differences seem to be constant, and we decide to calculate the mean error, splitting the snow depth values between

650 those greater than 0.3 $m$ and others.

Considering observed values more than 0.3 $m$ we found a mean error equal to 0.1218 $m$, whereas for the values greater than 0.3 $m$, we found a mean error of $0.1907m$.

Because our aim was to define a method to estimate snow depth which can be done automatically in the future without the longest procedure of visual estimation from images, we defined a corrected snow depth estimated time series in this way:

655  ● If $SD_{Sim}(t) < 30\ cm$: $SD_{Sim\ Cor}(t) = SD_{Sim}(t) + 0.1218\ m$;

  ● If $SD_{Sim}(t) > 30\ cm$: $SD_{Sim\ Cor}(t) = SD_{Sim}(t) + 0.1907m$;

Where $SD_{Sim}(t)$ is the original snow depth time series obtained by FMIPROT estimations and $SD_{Sim\ Cor}(t)$ is the corrected values obtained by adding a correction factor.

To verify visually the method's accuracy, in Figure 12, we reported the comparison between observed and simulated values

660 before and after correction. The new snow depth seems to be much closer to the observed one after this correction procedure. We calculated also RMSE and NSE before (BC) and after correction (AC). In case of considering the original snow depth estimation we obtained these accuracy parameters: $RMSE_{BC} = 0.1714\ m$, $NSE_{BC} =$0.6089. After the correction procedure we increased the simulations performances obtaining these accuracy values: $RMSE_{AC} = 0.05242\ m$, $NSE_{AC} =$0.9634. In addition, we have to highlight also that in this case the number of the algorithm failings (i.e., when snow depth is equal to NaN) is only

665 1 among 1054. So, the percentage is less than 0.1%.

Moreover, to test the model reliability, we selected images between 10 November 2018 and 3 May 2019. Unfortunately, this period is not optimal to test our model because most of the time snow depth on the ground was less than 0.1 $m$. Since we are interested to validate the method and check if the algorithm can retrieve with enough accuracy the snow depth, we referred only to the observed values greater than 0.1 $m$ .

670 With this aim, we ran FMIPROT, and using the correction factor previously defined, we compared the results with the observed snow depth, obtaining the graph reported in Figure 13. Results showed good accuracy with $RMSE = 0.06194\ m$ and $NSE = 0.7707$. These results are comparable to those reported for the period 2015-2017.



This case study was interesting because, even though the geometrical configuration wasn't optimal, most of the images were discarded and the high melting close to the stake occurred , studying in a critical way the results and simply applying a constant

correction factor, we can estimate snow depth values with enough accuracy.

### 4.3 CARESER DAM

The third case study is the Careser dam, in which we have only three visible images per day at 8:00 and 11:00 AM and 2:00 PM. In addition, in the last year, camera was broken, so we can't use images to retrieve the snow depth. For these reasons we selected the period between 1/1/2016 to 11/06/2018.

As in Gressoney La Trinitè Dejola case study, also here we cannot compare the FMIPROT algorithm estimations with manual or ultrasonic snow depth measurements. Thus, also in this case, we must refer to the visual estimation of images to validate our results. For the most of available images we estimated visually the snow depth by opening each image and writing in a txt files the height of the first visible marker above the snowpack layer. We use these measurements as a reference to compare the results. We have already said that this method is not optimal, and it's affected itself by an error which can be quantified in

$0.05\ m$ . Despite that Meteotrentino positioned this camera only for hydrological purposes, we obtained good results, with a simple camera and stake. We remarked also the changing in the camera in summer 2017, this is the reason why we have to choose different simulation parameters before and after that date. In fact, previous images have a worst pixel resolution, but the camera's view is much more focused on the stake. After summer 2017, we can observe a better pixel's resolution, but camera zoom decreased, and stake appeared much more far to the camera itself. In addition, we have to highlight the much

more camera's stability in the first period compared to the second one.
Initially we were worried about results because this site is located at an altitude near 2600 m, inside a prone windy area and with very bad climatic conditions in winter in terms of temperature and high snowfall. In addition, they collected images at 8:00, 11:00 AM and 2:00 PM, and especially in spring the last one is not optimal because high sunlight levels.
In Figure 14 we reported the comparison within the observation period: 1/01/2016-31/12/2018.

These results were obtained defining this set of parameters, for the period until June 2017: $L_S = 2\ m, T_S = 67, \sigma = 10$. In summer 2017, they changed the camera so we chose others set of parameters which can detect in the correct way the snow depth variability: $L_S = 2\ m, T_S = 68, \sigma = 3$.
The whole period of observation was reported in Figure 14. Finally, we calculated the accuracy indexes between visual inspection and FMIPROT estimations, obtaining these results:$RMSE = 0.108\ m, NSE = 0.916$.

In this case we can see that algorithm was very powerful, especially within the hydrological year 2016-17 in which snow depth was not so high and there were few snowfall events and a subsequent rapid melting. Despite that this site was not defined with the snow retrieval purpose, we obtained very good results in terms of accuracy. The only problem here, was the camera use limitation due to the bad atmospheric conditions which characterized a mountain region. If Meteotrentino will be interested in maintaining this system to snow depth retrieval, we suggest protecting camera against the strong windy conditions, building a

small wooden box, in which camera should be positioned. In addition, we have to highlight also that despite that the temporal resolution was poor, just 3 images per day, seems that we can follow the snowpack temporal variability with high accuracy. Probably, it would be better increasing the temporal resolution, which allows to have a snow depth time series with hourly resolution. Algorithm can detect well snow depth because marker stakes were larger than other case studies, in fact here stake had a width of $0.2\ m$. So, applying a gaussian filter parameter greater than 1 we can avoid misclassification, and correctly

detect stake's markers shape. We calculated the percentage of NaN over total images, which was equal to 1.5%.

### 4.4 CAUSES OF ESTIMATION FAILING

As shown in the previous paragraphs, especially when images had hourly resolution, in some cases this method cannot retrieve correctly the right snow depth value. Here we explain why could happen and in the next section how we can solve these





problems, improving the geometrical configuration. 8 Sources of algorithm failing were reported in Figure 15. founded in the three different case studies; this is the reason why we tested FMIPROT in different conditions, using different image resolutions, with different visibility conditions, luminance, brightness. Now we will try to describe each source of errors:

1) Shadows projected on the snowpack, which falls inside the ROI. In some cases, especially when the sun was behind the camera, and some buildings or an automatic weather station or trees, were located near the camera, shadows on the snowpack surface was detected as a marker by the algorithm picking phase. This happened in all the 3 case studies, but image n°1 explains the Sodankylä Peatland one, in which sunlight highlighted the automatic weather station located near the camera projecting its shape above the snowpack. This induce the reading of the shadow's position as a first visible marker above the ground, strongly underestimated the snow depth, that, in this case, must be near 1 m. To avoid this problem, we suggest removing all the possible sources of shadows near stake and camera.

2) Footprint, which alters the snowpack in front of the stake and appears inside the ROI. The same effect of that previously explained was caused by an alteration of the snowpack surface. Animals, men, or drops falling from the trees, can alter the surface, creating some discontinuities, which can be classified as a marker by the counting pixel algorithm phase. To avoid this problem, we suggest protecting the area close to the stake, making it inaccessible.

3) Obstruction of the camera's view. Especially in case of extreme snowfall events, snowflakes can partially or totally cover both camera's view or some part of the stake obscuring the first clean marker which must be visible to correctly detect the snow depth. Here the only suggestion is to protect the camera against rainfall and snowfall events, positioning it inside a box which can be made with wood or steel with one side open or with a transparent window (Plexiglas or glass ones).

4) Pixel reflectance alteration: Due to excessive exposure or direct sunlight in the camera's objective, some images, lost its true colors becoming unreadable. This problem occurs rarely, so it's difficult to find a geometrical design which can avoid this problem.

5) Rotation of the camera's view: extreme windy conditions can cause the camera's rotation, and stake which disappears from the camera's view. To maintain the camera in the same position, it must be positioned inside a wooden or steel box as previously explained in the point 3.

6) Blurred or opaque images: in some cases, can happen that pixels cannot be distinguish among others, so from the algorithm point of view markers appear as the same as the background, making impossible its counting.

7) Wind snow drift: strong wind velocity can carry snowflakes from the ground to the stake markers. In this case makers became white and indistinguishable from the white background.

8) Direct sunlight in front of the camera and high reflectance above the snowpack. In this case is not so clear why algorithm failed and what kind of protection has to be defined to avoid this problem.

**4.5 DESIGNING GEOMETRIC AND PARAMETERS CONFIGURATION**

Studying in detail results and causes of algorithm failing, we tried to define the proper geometrical configuration between camera and stake, highlighting the main characteristics which each single component must have:

- Relative position between camera and stake: the better configuration required that snow stake must be positioned as close as possible to the camera, so in this case the algorithm can detect more easily markers position because more than one pixel can discretize one single marker. The camera's height must be close to the middle of the stake's length trying to reduce the parallax effect: in this way we avoid the pixel size distortion. Between camera and stake it's better to remove all the possible obstructions such as branches of trees which can be identified as markers.

- Snow stake: we suggest a wooden stake, to reduce conductive heat transfer between stake and snowpack, limiting the sunlight reflectance on the snow. The background color must be yellow with black markers with $0.01\ m$ spacing. Stake width must be $0.2\ m$ and markers must cover half of that. It's better to make inaccessible the area near the





stake, to avoid problems of snowpack alteration caused by human or animal footprint. To increase stake stability, it's better to fix it on a steel plate which in turn can be fixed on the ground (preferable on the rock) with at least 4 expandable anchors sleeves with quick setting cement. The Ground or rocks surrounding the stake must be as flat as possible to avoid the gravitational movement of the first snowpack level, and the possibility of the horizontal translation of the stake.


- Camera: the pixel resolution is not so important if we respect an adequate distance from the stake, however we suggest using a reflex camera with the highest pixel resolution. It's mandatory to protect camera against wind, and precipitation events positioning it inside a wooden box of which one side must remain open or close with plastic or glass window.


- Connection with monitoring service: If characteristics of snowpack are required in real time there should be a permanent connection between the camera imaging storage and the monitoring service center. This can be power alimented thanks to a solar panel.

- Other instrumentations or field measurements: to cross correlate and validate algorithm estimations, periodically, it's mandatory compare algorithm estimations with manual measurements obtained by planned field campaigns.


Additionally we suggest to place: (a) an automatic weather station with temperature sensor for removing some misclassification (as example highest melting rate if temperature was not more than 0°C), and (b) an ultrasonic snow depth sensor, which can be used when algorithm cannot detect in the correct way snow depth values. The last suggestion can be that of positioning more than one stake inside the camera's view and compared the algorithm estimations, defining different region of interest selecting one by one each stake.

Generally, working with images to detect if one pixel could be classified as a snow covered one, we defined a brightness temperature of 127. But in the three case studies it was found that a brightness temperature threshold between 60 and 75 can be good to detect the snowpack layer (probably because we were interested more in the marker's shape detection despite the single pixel classification). In addition, it was found that in most of the cases a gaussian filter sigma of 1 was good to retrieve correctly markers shape and position. Moreover, to define a reliable snow depth time series it's better to repeat the analysis

with different parameters configuration and take the mean value among them to remove all possible errors which can affect a single time series as explained for the Sodankylä Peatland case study.

In addition, as the first image it is mandatory to have the whole snow stake not covered by snow, because we have to detect the right ROI (Region of Interest), which must have as the lower bound the ground level without snow. The highest contour level can be defined by the users remembering that it should be much more than the expected maximum snow depth in the

accumulation season. Consequently, and coherently with the previous phase, at the ROI identification it has to be associated the real stake length in m.





## 5. CONCLUSIONS

The main aim of this study was to explore the potentiality of time-lapse photography method to estimate the snow depth in various snow conditions. Three case studies were considered: one in Finland, and two in the Italian Alps. The first case study was properly designed for retrieving the snow depth by camera images. This was not the case for other two case studies in the Italians Alps where cameras were set up for other purposes (a simple monitoring of the snow depth on the ground which consists on estimate the snowpack depth with a visual inspection of images).

The results showed that, if the images were clear and the relative position between camera and stake allowed to detect stake's markers, the proposed method was successful to estimate the snow depth. In all the three case studies, the agreement between estimations and visual inspection of the images was evaluated with the Nash Sutcliffe Efficiency, which reached values greater than 0.9, both for Finnish and Italian sites.

To evaluate the accuracy, the Root Mean Square Error was calculated, which was higher in the Careser dam site than in other

two. This can be explained due to (1) the extreme weather conditions at 2600 m a.s.l. which may affect the camera's visibility and (2) the presence of the shadows above the snowpack layer, which can be classified as a stake's marker by the image's segmentation algorithm phase. There is a necessity to define the proper geometrical and parameter configuration for reducing the possible sources of estimation errors. I Camera must be positioned at the same distance from the stake's ends and stake must be a wooden one of 15 cm width, with a yellow background color and black markers which covered half of that. Camera

must be protected against extreme weather conditions (high wind velocity, intense rainfall or snowfall events) like positioning it inside a wooden or steel box. All possible sources of shadows must be avoided. In Sodankylä case study, we compared the performances of the proposed method with an ultrasonic snow depth sensor which measured the snow depth near the stake. In terms of NSE and RMSE, we found higher accuracy of the proposed method against the ultrasonic Campbell sensor, assuming manual measurements as reference.

In conclusion, we underline some strength points of the proposed method:

- High temporal resolution: depending on the camera's scan rate and storage data capability, we can build hourly or sub-hourly snow depth time series;
- High accuracy levels: if the geometrical configuration of camera-stake system and related infrastructures are perfectly designed, the proposed method estimates snow depth with accuracy levels comparable to the most commonly used
(manual measurements, ultrasonic/optical sensors);
- Reliable: percentages of algorithm failing are low and if some outliers are generated, we can easily detect and correct them with a post-processing procedure or simply opening the image and doing a "visual reading" of the first visible marker above the snow layer;
- Low cost: this can be considered as a low-cost solution with low maintenance costs;
- Remote sensing technique: can be easily extended in remote and dangerous areas such as mountain glaciers or polar regions in which currently there is a lack of data;
- Many other potential applications such as estimating snowfall events, pluviometer undercatch correction, snowfall and rainfall's splitting, estimating reservoir water levels or discharge or glaciers melting and monitoring in real-time lake or river's water levels in case of floods.

Our results indicates that this kind of technology has a good potential and it is a complementary solution to the satellite remote sensing for estimating snow depth automatically and remotely. Future developments are necessary, such as improving the algorithm for reducing possible misclassifications and errors and may be possible upscaling solutions.




**ACKNOWLEDGEMENTS**

About the Finnish case study, we thank to FMI (Finnish Meteorological Institute) in particular at Anna Kontu and Leena Leppanen for the snow depth measurement data series. About the Gressoney Dejola, we thank Daniele Catberro of the SMI
(Società Metereologica Italiana) for images and suggestions for data interpretations.

About the Careser dam case study, we thank Luca Froner who worked for Rete Sismica Trento and Civil Protection Agency of the Autonomous Province of Trento with webcam images.

**AUTHOR CONTRIBUTIONS**

Marco Bongio performed data analysis with advices from Ali Nadir Arslan and Carlo de Michele. The used retrieval algorithm on snow depth developed by Cemal Melih Tanis and implemented by Cemal Tanis and Marco Bongio under supervising by Ali Nadir Arslan and Carlo de Michele.  Marco Bongio wrote the manuscript with contributions from all co-authors.






**Tables and Figures**

| CASE STUDY | SENSOR | TIME PERIOD | RESOLUTION |
|---|---|---|---|
| **Sodankylä Peatland** | Campbell (Ultrasonic) | 15/10/2010-27/03/2019 | 10 minutes |
| | Manual Measurements | 09/10/2009-28/04/2015 | Weekly |
| | Camera Images | 06/11/2015-28/04/2019 | 30 Minutes |
| **Gressoney La Trinitè Dejola** | Visual Estimations (Aineva Operators) | 10/12/2013-20/05/2019 | 3 /Days |
| | Camera Images | 10/12/2013-20/05/2019 | Hourly |
| **Careser** | Camera Images | 01/01/2014- 31/12/2018 | 4/Days |

**Table 1. Data description related to each case study with time period and temporal resolution.**

| SNOW DEPTH ALGORITHM PARAMETERS | | | | |
|---|---|---|---|---|
| **Case Study** | **Simulation Period** | **ROI Length $L_s$ [m]** | **Threshold $T_s$** | **Gaussian Filter σ** |
| Sodankylä Peatland | 06/11/2015-28/04/2019 | 1.00 | 70 | 1 |
| Gressoney Dejola | 04/11/2014–31/12/2017 &10/11/2018-04/05/2019 | 1.20 or 0.3 | 70 | 4 |
| Careser Dam | 01/01/2016-26/05/2017 | 2.00 | 67 | 10 |
| Careser Dam | 04/11/2017-11/06/2018 | 2.00 | 68 | 3 |

**Table 2. Snow depth algorithm parameters related to each case study.**

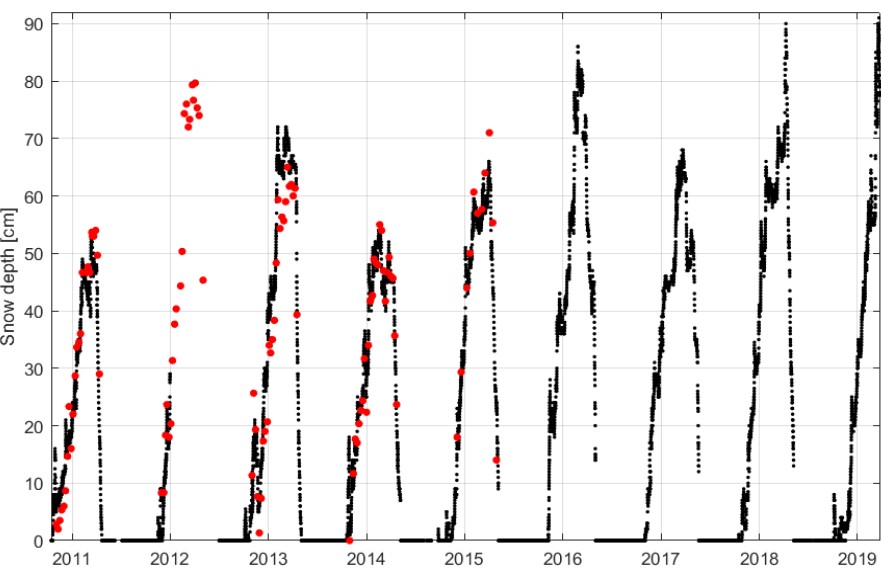

**Figure 1. Sodankylä Peatland snow depth in-situ Measurements: by Campbell ultrasonic snow depth sensor (black dots) and manual measurements (red dots).**





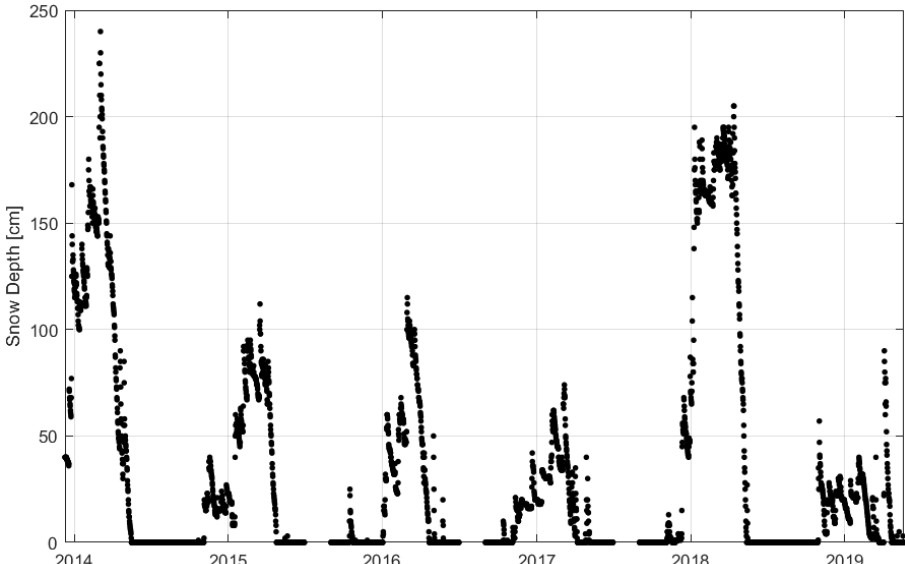

**Figure 2. Gressoney Dejola La Trinitè: snow depth visual estimations carried out by AINEVA operators, in the period 2013-2019.**


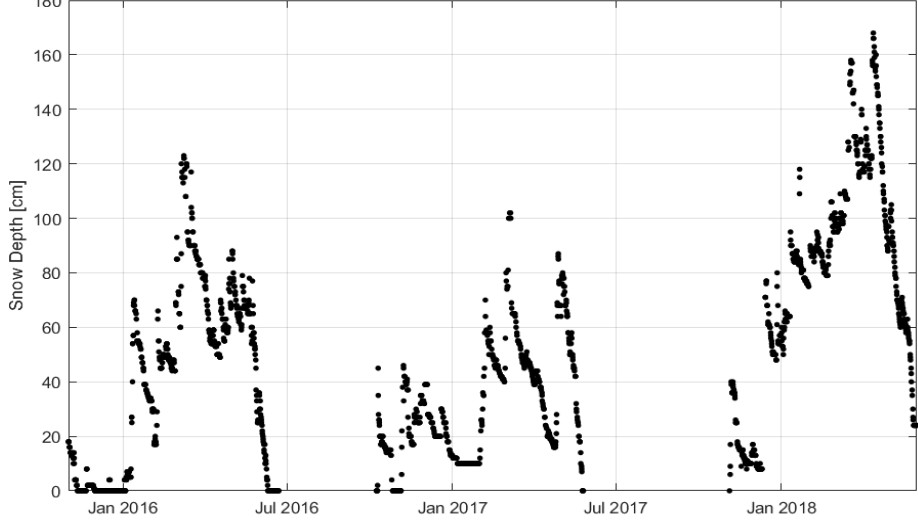

**Figure 3. Careser dam: Snow depth visual estimations between 1st November 2015 to 31th December 2018.**



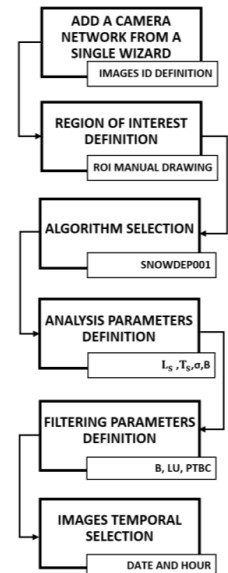

**Figure 4. Snow depth retrieval procedure: FMIPROT flow chart.**

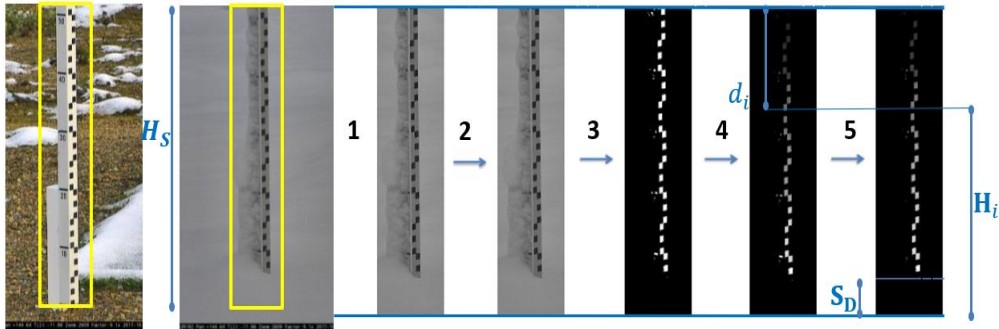


**Figure 5. Snow depth estimation flow chart algorithm (Spice Site - Sodankylä).**



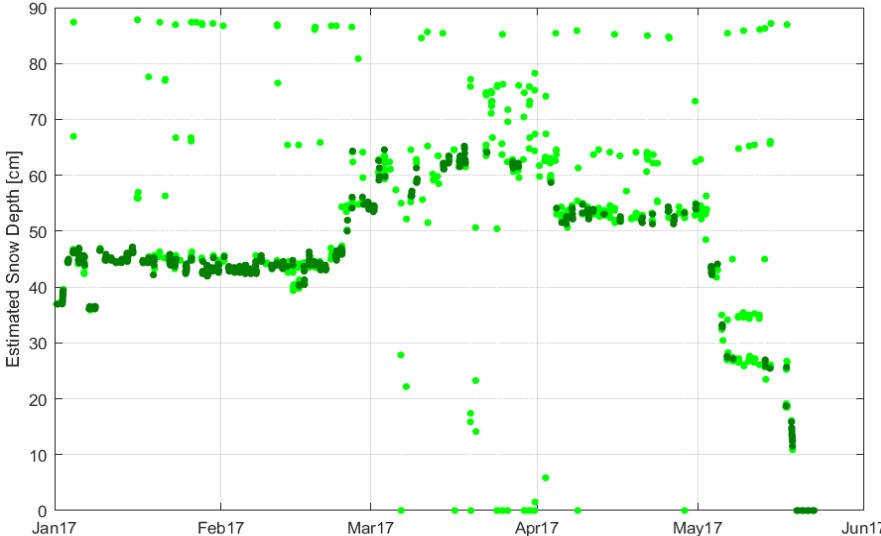

**Figure 6. Sodankylä Peatland: Smoothing algorithm correction of FMIPROT applied to the snow depth estimate in the first half of 2017. In light green dots we reported the snow depth originally estimated by FMIPROT using $Ts = 70$ and $\sigma = 5$, with dark green dots the corrected snow depth time series.**


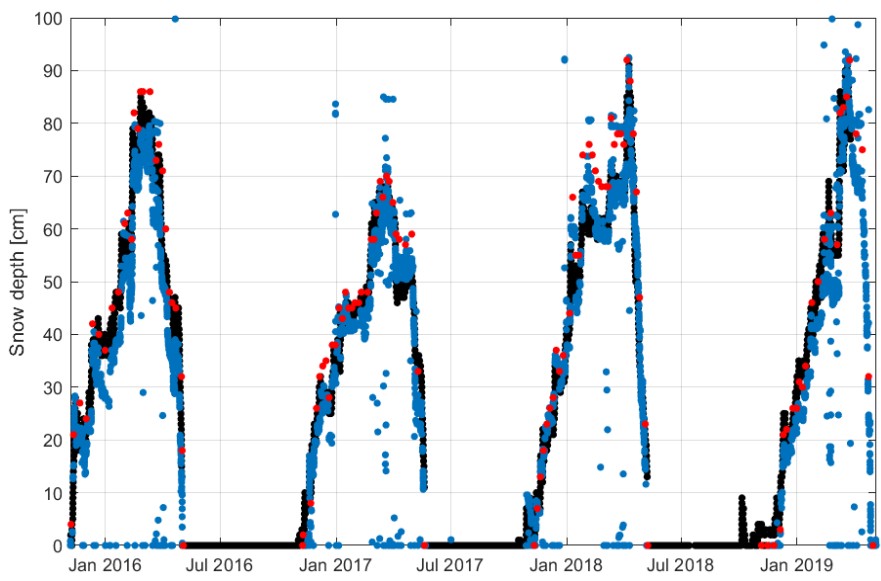


**Figure 7. Sodankylä Peatland: Snow depth estimations comparison: Ultrasonic measurements (black dots), estimated by FMIPROT (blue dots), visual observations (red dots).**



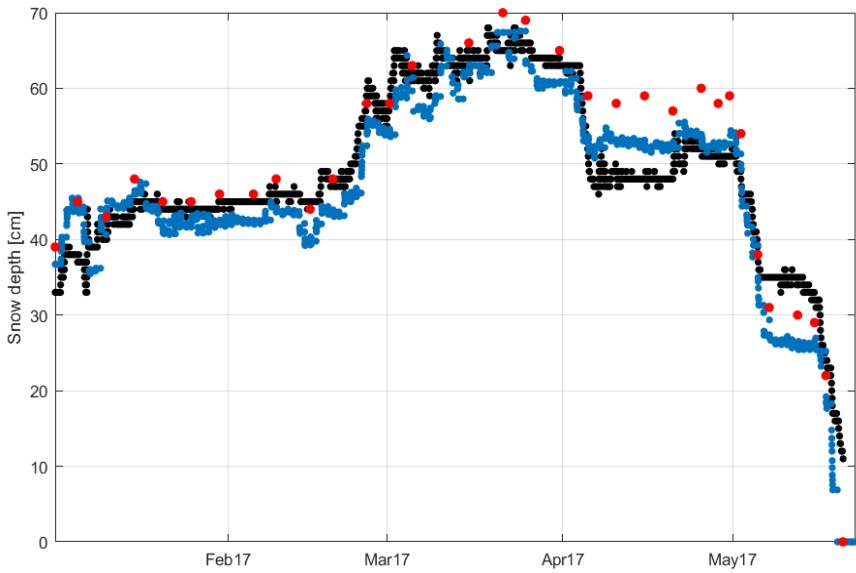


**Figure 8. Sodankylä Peatland: Comparison between visual estimations (red dots), ultrasonic measurements (black dots), FMIPROT estimations (blue dots) in the first half of 2017.**

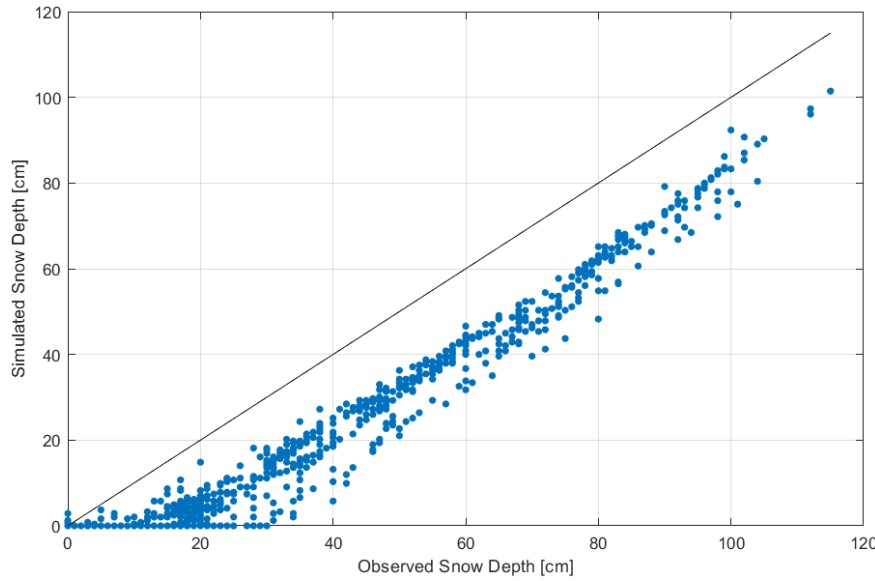

**Figure 9. Gressoney La Trinitè Dejola: Scatterplot between estimated by FMIPROT (before correction procedure) and observed**


**values.**



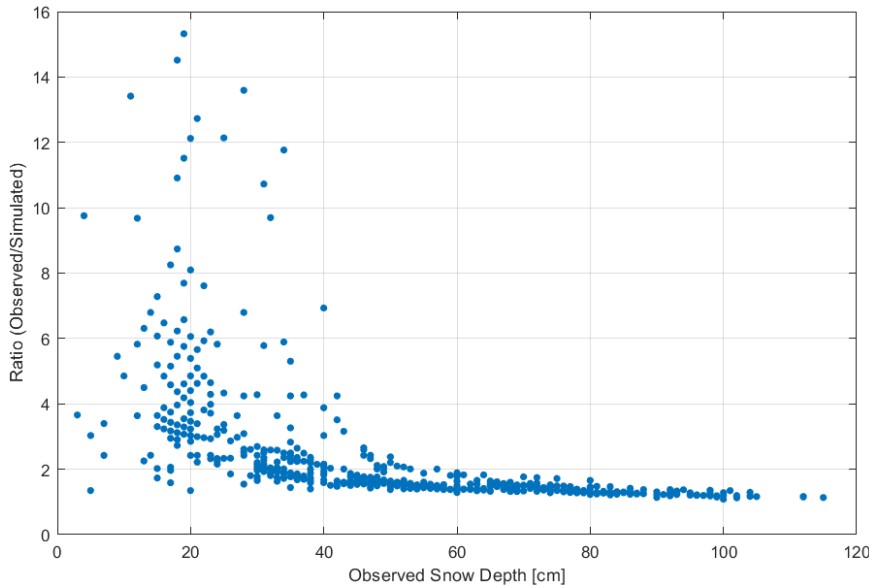

**Figure 10. Gressoney La Trinitè Dejola: Ratio between observed and estimated snow depth values in the calibration period.**

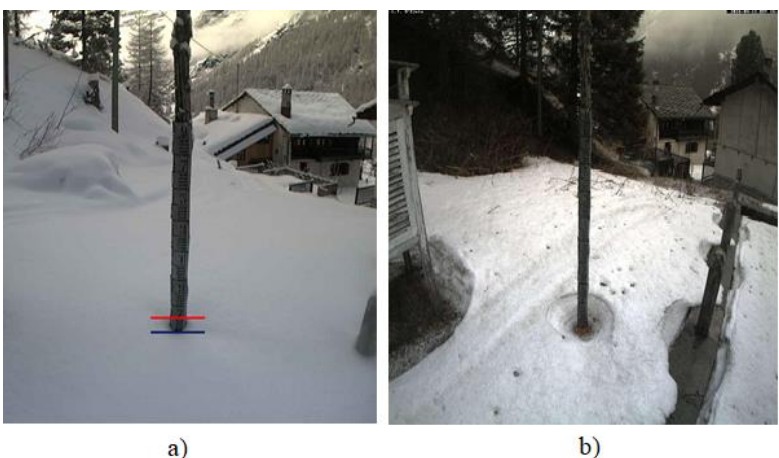

**Figure 11. a) Images from Gressoney Dejola camera, taken at 8 AM of 09/02/2017. In red we marked the AINEVA Operator estimation, in blue the first selected marker from FMIPROT Algorithm. b) 13/04/2016 Image from Gressoney la Trinitè Dejola camera which shows the high melting rate close to the stake.**



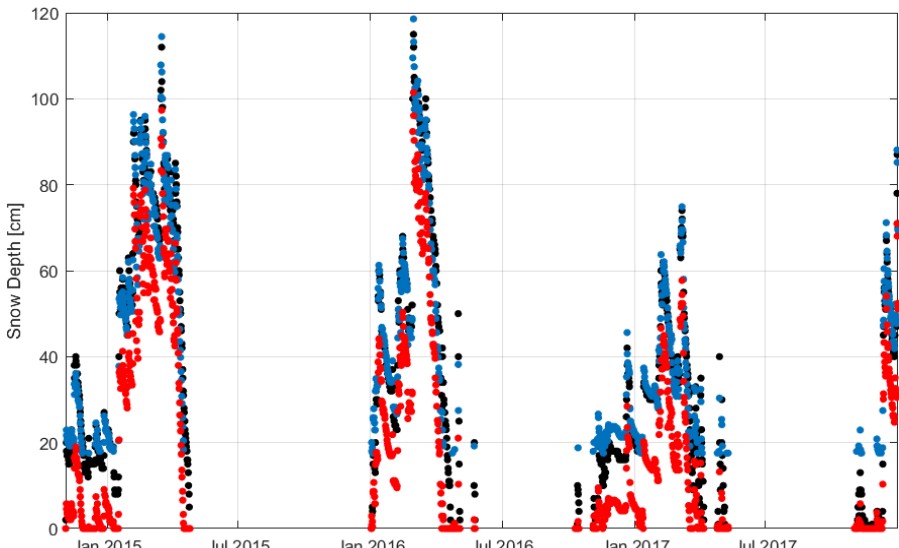

**Figure 12. Gressoney La Trinitè Dejola case study: comparison between visual estimations (black dots) and estimated snow depth after (blue dots) and before (red dots) correction, in the calibration period.**


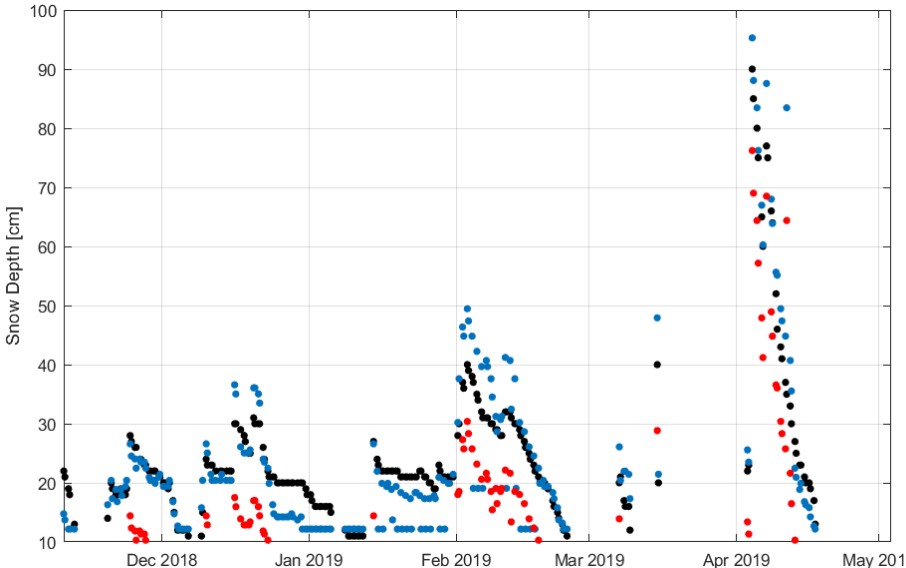

**Figure 13. Gressoney La Trinitè Dejola: comparison between observed (black dots) and simulated snow depth after (blue dots) and before (red dots) correction, in the validation period (11/2018-05/2019).**






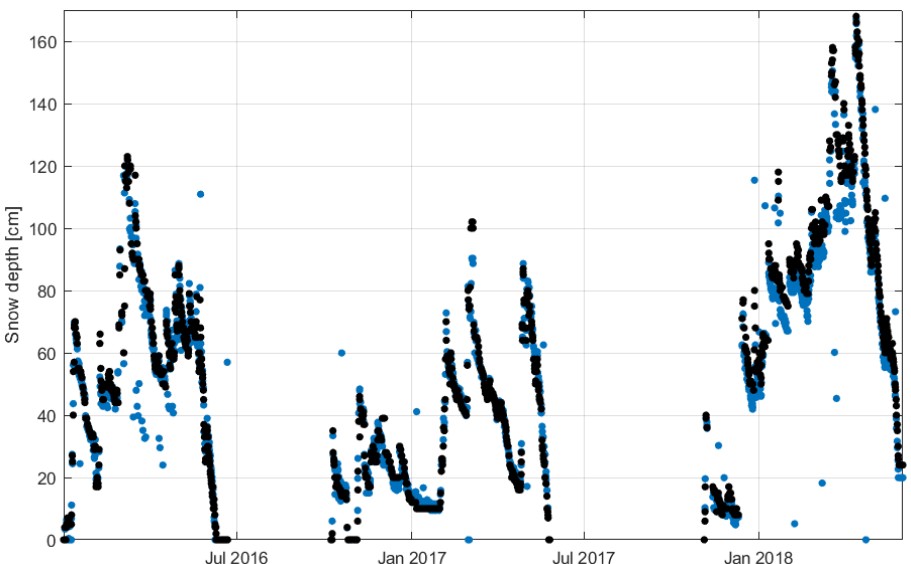

**Figure 14. Careser dam case study. Comparison between snow depth retrieved by FMIPROT (blue dots) and visual estimation of images (black dots) for the period 1/1/2016-01/06/2018.**


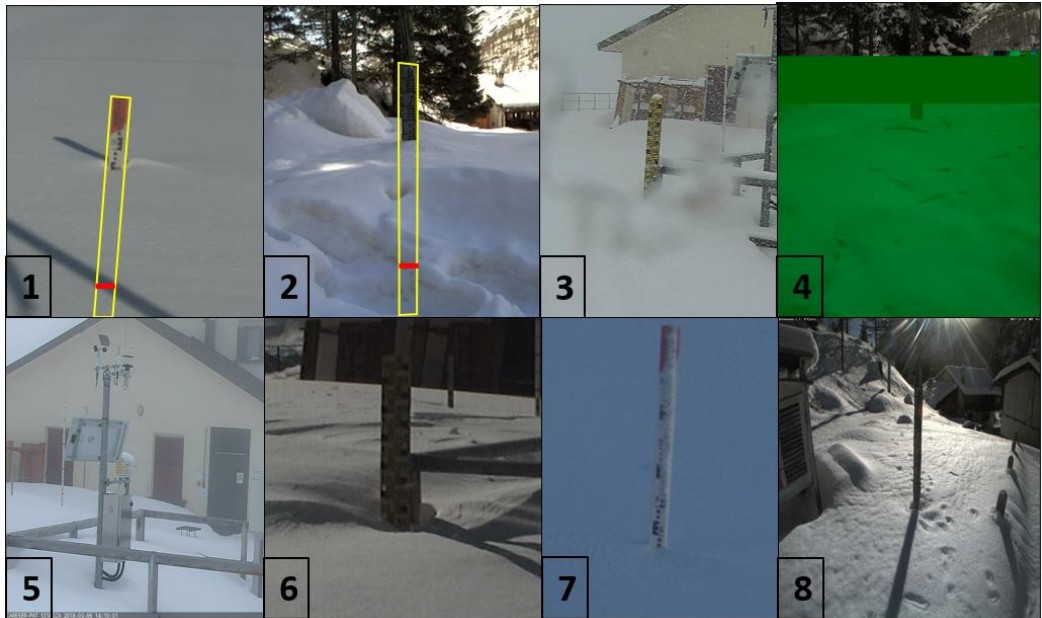

**Figure 15. Sources of algorithm failing.**




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
