# Peer review of "Snow depth time series retrieval by time-lapse photography: Finnish"

_The Cryosphere, 2019_

## Short Comment (SC1) · 7 Oct 2019

Due to some updates and modifications going on in the webpage system, the URL http://fmiprot.fmi.fi is not available at the moment. The page is hosted only at https://fmiprot.fmi.fi until the modifications are done.

---

## Referee Comment (RC1) · Anonymous Referee #1 · 14 Nov 2019

The authors describe in the manuscript an automatic procedure aimed to estimate the snow depth from stake positioned in front of a camera. They considered three different sites and two different approaches in order to estimate the efficiency of their tool. Results are promising but I suggest to fix some issues in order to improve the readability of the paper.

1) too long introduction - authors are focusing the attention on snow depth and they should refer strictly on SD and secondly on the relative impact on other parameters - authors describe methods for estimating SD using about 40 lines but the considered methods required less 10 lines. Are satellite and airborne approaches useful in this manuscript? You considered only ultrasonic tools and visual inspection of camera pictures...

Interactive
comment

2) case studies - You should highlight better that in Sodankyla you considered camera detection (automatic and manned) versus ultrasonic estimations and that the snow pit observations confirm or not the ultrasonic "ground-thruth". - What is the distance between camera and the 3 stakes? - What is the Camera height? - What is the distance between the ultrasonic device and the camera? - How many picture you have acquired and how many can be used? - How many ultrasonic values you have in the same period?

- Gressoney photo – manual inspection - What is the distance between camera and stake? - What is the Camera height? - How many picture you have acquired and how many can be used?

- Careser dam photo – manual inspection - What is the distance between camera and stake? - What is the Camera height? - How many picture you have acquired and how many can be used?

- Authors should discuss how many valid observations they have obtained compared an ultrasonic device before extrapolating time-series.

You showed for Sodankyla time-series in Fig.7 and 8 without showing correlation plots between ultrasonic, manned and automatic estimations. I see several strong understimations while the snow depth is very high. How many? Why?

You showed for Gressoney time-series in Fig.9 showing the correlation plots between manned and automatic estimations. What do you mean with "simulated"? You defined two different correction factors, are they site-specific or operator-specific?

You showed for Careser dam time-series in Fig.14 without showing the correlation plots between manned and automatic estimations.

- How much the distance between the camera and the stack affects the determination? - They are discussing about an optimal geometry (4.5): could you give distances or heights? Can you suggest a final setup and discuss much effort require this kind of
facility compared to an ultrasonic device?

Minor issues: - Numbers...10*10ˆ-2m or 0.01m - several font bugs

---

## Referee Comment (RC2) · Anonymous Referee #2 · 5 Mar 2020

General comments

This study presents application of FMIPROT tool to estimate snow depth from images of snow stakes at selected sites in Finland and Italy.

In my opinion, topic is interesting and within the scope of the journal. However, the manuscript is, in my opinion, not ready for publication in its current form. The main reasons are not clearly formulated and demonstrated the novel scientific contributions of the study. The Introduction section is rather general and does not clearly present what the current status of the snow depth retrieval by digital camera is and what the recent research gaps are. The list of innovative aspects of the works then include large number of points, but the novelty of many of them is rather low or not clear (these are not clearly connected with the results of previous studies). So I would suggest to make

the Introduction section and formulation of the aims and novelty much more targeted and specific.

If the main objectives are to evaluate the tool, then I would suggest to present methodology/tools first and then describe the dataset used in this study. The description of the sites can be shortened and information which is relevant for all tested sites can be mentioned only once (for example information that in the late afternoon the images are very dark is mentioned for different sites, but it will be enough to indicate that once for all sites and specific times for specific sites can be indicated for example in a table). The Method section reads like a manual to the tool, but this needs to be clearly linked with the main objectives of the paper and formulation/demonstration of the novelty. So please consider to revise this part and describe methodological steps/approaches which are considered here as a novel contribution. The same applies for Results. Please present here more clearly some story and take home message for the readers, i.e. what the new findings are. Finally I missed a discussion section which can link (and compare) the new findings of this study with previous research. This part can also include some lessons learned part and implications of this study for the future investigations.

---

## Referee Comment (RC3) · Anonymous Referee #3 · 18 Mar 2020

"The manuscript Snow depth estimation by time-lapse photography: Finnish and Italian case studies" by Bongio et al. attempts at reporting a new algorithmic method to derive snow depth from time-lapse images. This topic has been covered in many studies for many years, some of them even referenced by the authors. In the current form of the manuscript, it is really hard to see what this study really brings to complement previous studies. As a simple example, it lacks a simple comparative tables of performance of this "new" method with older ones. But even besides the content, this manuscript reads like a poorly written report. Too many sentences have syntax problems. And even more importantly, this manuscript requires serious restructuring and refocusing to be qualifiable as a scientific manuscript. It will require a clear motivation of the work, a clear background section that allow the reader to understand specifically what this new

study brings in comparison to previous work.

Within the very first sentence of the abstract there are issues. See below: line 9: potentiality could be replaced by potential line 9: "boreal forested", here it should read " the boreal forest" or in the "forested boreal regions" line 10: what is meant by snowboards? By default this refers to the gliding object on snow unless further specified. line 14: What is an "ad-hoc" algorithm. The combination of ad-hoc and quotes is really unclear. And what does this mean? line 19: "correction the well known ..." this is grammatically incorrect, the whole sentence has syntax issues.

The introduction is equally problematic:

Line 30: why is "Earth Energy balance" with capital letters? Line 36: this sentence is overly complicated for a very simple meaning. Line 46: you mention "a lot of studies" but cite only one . . .

All the way to line 105 (introduction section) we have no idea what the manuscript is about. There is a lot of background on snow hydrology and observation techniques with no motivation as to why it is relevant to the manuscript. Moreover each method to observe snow depth is described as "possible" with its limitation, which reads as an unfair trial to all of them.

Finally, the sentence "In these studies, the principal aim is to show how time-lapse photography could be used for investigating snow processes, but they were not properly focused in snow depth retrieval purposes." is vague and unfounded. What is meant by the term "properly"? Almost all of these studies have a primary focus at retrieving snow depth from time lapse imagery.

Overall, this manuscript would require major rewriting before a possible evaluation of its content.

---

## Author Comment (AC1) · 8 Apr 2020

We would like to thank the Reviewer for the time and valuable comments to our work. Below we will report our replies to each of your comments following this format:

Reviewer'1 comment

Authors reply

Reviewer #1 says "The authors describe in the manuscript an automatic procedure aimed to estimate the snow depth from stake positioned in front of a camera. They considered three different sites and two different approaches in order to estimate the efficiency of their tool. Results are promising but I suggest to fix some issues in order to improve the readability of the paper."

[Figure]

Authors reply: Many thanks for your kind comment.

Reviewer #1 says: "1) too long introduction - authors are focusing the attention on snow depth and they should refer strictly on SD and secondly on the relative impact on other parameters - authors describe methods for estimating SD using about 40 lines but the considered methods required less 10 lines. Are satellite and airborne approaches useful in this manuscript? You considered only ultrasonic tools and visual inspection of camera pictures..."

Authors reply: In the revised version of the manuscript, we have condensed the introduction, we focused on snow depth, and time-lapse photography.

Reviewer #1 says: "2) case studies - You should highlight better that in Sodankyla you considered camera detection (automatic and manned) versus ultrasonic estimations and that the snow pit observations confirm or not the ultrasonic "ground-thruth".

Authors reply: Yes. Sodankyla case study is the only one in which we can compare time-lapse photography (FMIPROT) retrievals versus ultrasonic measurements, so, first we tried to check if ultrasonic measurements and manned ones are comparable in an overlapping period of observation, defined as "calibration". This fact allows to use the ultrasonic measurements as "the ground truth" to compare, in a different time period, ultrasonic sensor and FMIPROT retrievals.

Reviewer #1 says: "What is the distance between camera and the 3 stakes? - What is the Camera height? - What is the distance between the ultrasonic device and the camera? - How many pictures you have acquired and how many can be used? - How many ultrasonic values you have in the same period?"

Authors reply: The camera features are: - Distance between camera and stakes: from the left to the right respectively: 10, 20.2 and 4.16 m. The camera height is 2.08 m above the ground with a vertical angle of 18° (pitch) and horizontal one of -2° (roll). - The ultrasonic sensor is at the SO003 station, 222 m far from the Peatland site.

The available dataset related to this site is:

- Campbell (Ultrasonic Sensor): from 03/11/2011 to 27/03/2019; n° of measurements: 383963; temporal resolution:10 minutes; - Manual Measurements: from 03/11/2011 to 28/04/2015; n° of measurements: 105; temporal resolution:Weekly; - Visual Estimations: from 07/11/2015 to 01/05/2019; n° of measurements: 129; temporal resolution:Daily; - Camera Images: from 06/11/2015 to 28/04/2019; n° of measurements: 11604; temporal resolution:30 Minutes;

Ultrasonic measurements had a resolution of 10 minutes and didn't present noise, so probably were published after a correction procedure. We used 11604 images and selected the period from 06/11/2015 to 28/04/2019 from 9.00 AM to 3.00 PM because: - Before 9.00 AM and after 3 PM images were in most of the cases totally black; - The relative position between camera and stake was the same;

We have added this info in the revised version of the manuscript.

Reviewer #1 says: "- Gressoney photo – manual inspection - What is the distance between camera and stake? - What is the Camera height? - How many pictures you have acquired and how many can be used?"

Authors reply: The features of this camera-stake configuration are: - Camera's height: 2 m; - Distance between camera and stake: 10 m; - Type of camera: Siap Micros without protection.

Related to the images: AINEVA operator gave us images at hourly resolution from :2014-2019, generally from 6.00 AM to 8 PM. However, we decided to use only images from 01/11/2014 to 31/12/2017 and 1/11/2018 to 01/05/2019, at 8.00 am and 2.00 PM because in this period we have these 2 mandatory conditions: 1) AINEVA operators gave us an excel file in which they reported the snow depth visual estimated. With this information we could compare our results; 2) The relative position between camera and stake wasn't changed; Within this period, we had and we used for our estimations

1058 images. We have added this info in the revised version of the manuscript.

Reviewer #1 says: "- Careser dam photo – manual inspection - What is the distance between camera and stake? - What is the Camera height? - How many pictures you have acquired and how many can be used?"

Authors reply: The features of this camera-stake configuration are: - Camera's height: 3 m; - Distance between camera and stake: 6 m; - Type of camera: Axis 214 PTZ with semispherical pet protection. We have added this info in the revised version of the manuscript.

About images: the Civil Protection Agency of Trentino gave us images from 01/01/2014 from 31/12/2018, 4 images per day at 5:15,8:15,11:15 AM and 2:15 PM. As in the previous case, we decided to remove the first (at 5:15) because in most of the cases were totally black, and we selected the period from 01-01-2016 to 01-06-2018, in which the relative position between camera and stake was the same. We used 1714 images.

Reviewer #1 says: "- Authors should discuss how many valid observations they have obtained compared an ultrasonic device before extrapolating time-series."

Authors reply: In order to validate the reliability of the ultrasonic snow depth sensor, we compared manual measurements with the ultrasonic data. These two datasets have following features: - Ultrasonic measurements: available for the period 03/11/2011 to 27/03/2019 every ten minutes (383963 observations). In the graph we plotted these measurements at hourly resolution; - Manual measurements: available for the period 03/11/2011 to 28/04/2015, with a daily resolution. During this period the manual measurements were not conducted every day. There was a lack of data in comparison with the ultrasonic measurements. These snow depth time series (105 manual observations) were measured by rulers by FMI researchers, generally in the morning; So, the comparison between the manual measurements and the ultrasonic measurements was done for 105 observations which were measured at the same hour/day/month/year.

In the revised version of the manuscript, we have discussed this issue.

Reviewer #1 says: "You showed for Sodankyla time-series in Fig.7 and 8 without showing correlation plots between ultrasonic, manned and automatic estimations. I see several strong understimations while the snow depth is very high. How many? Why?"

Authors reply: Unfortunately, we cannot compare all these three methods because manual measurements are not available when the camera was installed. High underestimations occurred simply because when the algorithm fails and cannot identify correctly the snow depth level, it gives zero as value. This happened in some cases (obstruction of the camera's view, poor visibility conditions...) now reported in the "discussion" section of the revised manuscript.

The correlation plot, in the Gressoney case study helped us to define the method to correct the data, trying to minimize FMIPROT estimation errors. In the Sodankyla case study, we tried to correct measurements with another approach that is those reported in the figure S4 in the supplement document, defining an ensemble of simulations with different parameters. In this way we showed two different solutions to make our results more affordable.

Reviewer #1 says "You showed for Gressoney time-series in Fig.9 showing the correlation plots between manned and automatic estimations. What do you mean with "simulated?"

Authors reply: We mean "retrieved" by FMIPROT. We have fixed it in the revised version of the manuscript.

Reviewer #1 says: "You defined two different correction factors, are they site-specific or operator-specific?"

Authors reply: Operator specific. We have fixed it in the revised version of the manuscript.

Reviewer #1 says: "You showed for Careser dam time-series in Fig.14 without showing

the correlation plots between manned and automatic estimations. - How much the distance between the camera and the stack affects the determination?

Authors reply: We have not investigated how much the distance between camera and stake could affect our measurements. For sure, increasing the distance will increase the possibility that some obstructions or some shadows appear on the snowpack and so the algorithm could detect those as snow levels. Furthermore, if the stake 1 were positioned at a greater distance than the stake 2, in the former case, within a single pixel we could observe 2 black markers, while in the latter case only one. Therefore, in the first case, the algorithm is unable to distinguish between the two markers, reflecting in a poor retrieval of the snow depth.

Reviewer #1 says: "They are discussing about an optimal geometry (4.5): could you give distances or heights? Can you suggest a final setup and discuss much effort require this kind of facility compared to an ultrasonic device?"

Authors reply: It's difficult to say how it is the optimal geometric configuration, because it depends strongly on the camera's parameter, type of objective, lenses. Of course, a rough suggestion can be given as: a stake 2 m long, a camera-stake distance of 4-5 m, and a camera height of 1.5 m. We clarified this in the revised version of the manuscript.

Reviewer #1 says: "Minor issues: - Numbers...10*10Ȩ̈Ę-2m or 0.01m - several font bugs"

Authors reply: Fixed

---

## Author Comment (AC2) · 8 Apr 2020

We would like to thank the Reviewer for the time and valuable comments to our work. Below we will report our replies to each of your comments following this format:

Reviewer'2 comment

Authors reply

Reviewer #2 says :"This study presents application of FMIPROT tool to estimate snow depth from images of snow stakes at selected sites in Finland and Italy. In my opinion, topic is interesting and within the scope of the journal. However, the manuscript is, in my opinion, not ready for publication in its current form. The main reasons are not clearly formulated and demonstrated the novel scientific contributions of the study."
Authors reply: In the revised version of the manuscript, we made major modifications in order to formulate and demonstrate our novel scientific contributions. In few words, we introduced a new methodology which is, totally automatic and easy to use for retrieving snow depth. We demonstrated our methodology successfully in 3 different case studies. Our methodology will be very useful for many applications including scientific research where the snow depth retrieval is needed where the AWS station installation is too dangerous or impossible. In the manuscript, we also showed that our methodology on the retrieval of snow depth can be more practical and accurate at the test sites, where configurations such as stake and camera positions were not planned, just using the existing configurations.

Reviewer #2 says :"The Introduction section is rather general and does not clearly present what the current status of the snow depth retrieval by digital camera is and what the recent research gaps are."

Authors reply: We modified the introduction section where the current status of snow depth retrieval by digital camera and recent research on snow depth retrieval and gaps are clearly visible.

Reviewer #2 says :"The list of innovative aspects of the works then include large number of points, but the novelty of many of them is rather low or not clear (these are not clearly connected with the results of previous studies). So, I would suggest to make the Introduction section and formulation of the aims and novelty much more targeted and specific."

Authors reply: Yes, we have re-formulated the Introduction section, focusing the attention to time-lapse photography and the open questions.

Reviewer #2 says :"If the main objectives are to evaluate the tool, then I would suggest to present methodology/tools first and then describe the dataset used in this study."

Authors reply: Our objective is to present a new methodology which is totally automatic

and easy to use for retrieving snow depth. We modified the manuscript as suggested.

Reviewer #2 says :"The description of the sites can be shortened and information which is relevant for all tested sites can be mentioned only once (for example information that in the late afternoon the images are very dark is mentioned for different sites, but it will be enough to indicate that once for all sites and specific times for specific sites can be indicated for example in a table)."

Authors reply: Yes, in the revised version of the manuscript, we have shortened the description of case studies.

Reviewer #2 says :"The Method section reads like a manual to the tool, but this needs to be clearly linked with the main objectives of the paper and formulation/demonstration of the novelty. So please consider to revise this part and describe methodological steps/approaches which are considered here as a novel contribution. The same applies for Results. Please present here more clearly some story and take home message for the readers, i.e. what the new findings are."

Authors reply: Yes, in the revised version of the manuscript, we have reformulated the methodology into more scientific presentation. We also rewrote the results more clearly underlining take home messages and new findings for readers.

Reviewer #2 says :"Finally I missed a discussion section which can link (and compare) the new findings of this study with previous research. This part can also include some lessons learned part and implications of this study for the future investigations."

Authors reply: In the revised version of the manuscript, we have distinguished "Results" section from "Discussion" where we report lessons learned and implications for the future investigations.

---

## Author Comment (AC3) · 8 Apr 2020

We would like to thank the Reviewer for the time and valuable comments to our work. Below we will report our replies to each of your comments following this format:

Reviewer's 3 comment

Authors reply

Reviewer #3 says: "The manuscript Snow depth estimation by time-lapse photography: Finnish and Italian case studies" by Bongio et al. attempts at reporting a new algorithmic method to derive snow depth from time-lapse images. This topic has been covered in many studies for many years, some of them even referenced by the authors. In the current form of the manuscript, it is really hard to see what this study really brings to

complement previous studies."

Authors reply: In the revised version of the manuscript, we have improved its readability.

Reviewer #3 says: "As a simple example, it lacks a simple comparative tables of performance of this "new" method with older ones. But even besides the content, this manuscript reads like a poorly written report. Too many sentences have syntax problems. And even more importantly, this manuscript requires serious restructuring and refocusing to be qualifiable as a scientific manuscript."

Authors reply: In the revised version of the manuscript, we have heavily restructured and refocused the manuscript, as follows:

- Shortened and focused the "Introduction" section, - Reshaped the "Methodology" section, - Reported first "Methodology" and then "Case studies", - Shortened the "Case study" description, - Separate sections for "Results" and "Discussion"

Reviewer #3 says: "It will require a clear motivation of the work, a clear background section that allow the reader to understand specifically what this new study brings in comparison to previous work."

Authors reply: Yes, we agree. We have specified this in the revised version of the "Introduction" section.

Reviewer #3 says: "Within the very first sentence of the abstract there are issues. See below: line 9: potentiality could be replaced by potential line 9: "boreal forested", here it should read " the boreal forest" or in the "forested boreal regions" line 10: what is meant by snowboards? By default this refers to the gliding object on snow unless further specified. line 14: What is an "ad-hoc" algorithm. The combination of ad-hoc and quotes is really unclear. And what does this mean? line 19: "correction the well known ..." this is grammatically incorrect, the whole sentence has syntax issues."

Authors reply: Fixed, not only in the Abstract, but throughout all the manuscript.
Reviewer #3 says: "The introduction is equally problematic: Line 30: why is "Earth Energy balance" with capital letters? Line 36: this sentence is overly complicated for a very simple meaning. Line 46: you mention "a lot of studies" but cite only one : : : All the way to line 105 (introduction section) we have no idea what the manuscript is about. There is a lot of background on snow hydrology and observation techniques with no motivation as to why it is relevant to the manuscript. Moreover each method to observe snow depth is described as "possible" with its limitation, which reads as an unfair trial to all of them. Finally, the sentence "In these studies, the principal aim is to show how time-lapse photography could be used for investigating snow processes, but they were not properly focused in snow depth retrieval purposes." is vague and unfounded. What is meant by the term "properly"? Almost all of these studies have a primary focus at retrieving snow depth from time lapse imagery."

Authors reply: We agree and the "Introduction" section has been heavily modified, shortened and focused on the time-lapse photography.

Reviewer #3 says: "Overall, this manuscript would require major rewriting before a possible evaluation of its content."

Authors reply: We really hope that the revised manuscript with all modifications will answers the Reviewer's comments.

---

## Author Response (AR1)

**AUTHOR'S RESPONSE**

In the Manuscript you can find the new version, with highlighted, all the major modifications.

Here we reported a list of the section describing principal improvings inside of those:

**GENERAL DESCRIPTION:**

- We made major modifications in order to formulate and demonstrate our novel scientific contributions. We introduce a new methodology which is, totally automatic and easy to use for retrieving snow depth. We demonstrated our methodology successfully in 3 different case studies. Our methodology will be very useful for many applications including scientific research where the snow depth retrieval is needed where the AWS station installation is too dangerous or impossible. In this manuscript, we also showed that our methodology on the retrieval of snow depth can be more practical and accurate at the test sites, where configurations such as stake and camera positions were not planned, just using the existing configurations.

    - Shortened and focused the "Introduction" section,

    - Reshaped the "Methodology" section,

    - Reported first "Methodology" and then "Case studies",

    - Shortened the "Case study" description,

    - Separate sections for "Results" and "Discussion"

- In the revised version of the manuscript, we have improved its readability.

**INTRODUCTION:**

We rewrote this part taken into considerations all reviewers' comments:

- In the revised version of the manuscript, we have condensed the introduction, we focused on snow depth, and time-lapse photography.

- We modified the introduction section where the current status of snow depth retrieval by digital camera and recent research on snow depth retrieval and gaps are clearly visible.

- We have re-formulated the Introduction section, focusing the attention to time-lapse photography and the open questions.

**METHOD:**

We reorganized and structured Method chapter with the comments from reviewers.

- In the revised version of the manuscript, we have reformulated the methodology into more scientific presentation. We also rewrote the results more clearly underlining take home messages and new findings for readers.

- We have removed the subchapter "FMIPROT" and mentioned it in the chapter in the beginning.

- The procedure on how to use FMIPROT to estimate the snow depth is removed, as it does not provide any scientific value, but rather shows the usage of the software.

**CASE STUDIES**

We reorganized and restructured this chapter with the comments from reviewers.

We have shortened the description of case studies. We have used cm rather than 10^x m, throughout the manuscript.

**RESULTS**

We reorganized and restructred this chapter with the comments from reviewers.

- Subchapters of causes of failures and configuration design suggestions are moved to the new chapter, Discussion.

40

- Relevant items in the chapter are moved to the new chapter, Discussion.

**DISCUSSION**

New chapter added based the comments from reviewers.

In the revised version of the manuscript, we have distinguished "Results" section from "Discussion" where we report lessons learned and implications for the future investigations.

---

## Author Response (AR2)

**COVER LETTER**

We would like to thank all reviewers for their valuable comments and inputs. They surely improved our manuscript. We modified the manuscript accordingly to your suggestions.

In the following we will show:

- Manuscript with text changes highlighted in yellow colors;
- Answers to anonymous referee 1: our replies were written in light blue color following reviewers comments/questions/suggestions;
- Answers to anonymous referee 2: our replies were written in light blue color following reviewers comments/questions/suggestions;

We hope that our answers and updates will be satisfactory and looking forward to hearing outcome of this minor revision.

With Best Regards,

Marco Bongio

on behalf of the other authors.

[revised manuscript text omitted]

**Answers to Anonymous Referee 1**

**Abstract**

Lines 22-25 the NSE is not a so diffused statistical index as the RMSE index. The not-expert reader cannot interpret numbers in the abstract without a detailed knowledge of mentioned indexes.

Thanks for this comment. We agree that this index isn't so diffused as the RMSE but, the Cryosphere is a scientific journal, all of Readers probably have the knowledge to understand what this index means.

**Introduction**

There are a lot of references but there is also a critical inconsistency: the manuscript is focused on snow depth (looking at the title), good to say that SD is important for estimating SWE from eq.1 (maybe "not rather than" as stated in lines 35-36) and how that it is possible to measure it (lines 40-49). Authors define accuracy for different reference methods focused on SD in lines 45-49 (probably from Pirazzini et al 2018). This is the most important statement in the introduction since the reader should understand which is the state of the art about measuring the SD. From line 50 to 55 authors summarize the state of the art about time-lapse photography (good to do that in this case) stating "In these studies...but they were not specifically focused on the retrieval of snow depth". From line 56 to 114 they try to extract information about estimating SD from some of those papers: no information about SD in lines 56-70 and 82-85; 10cm accuracy in Parajka et al 2012 (lines 71-81); 10cm accuracy and 7cm RMSE? Against visual inspection in Garvelmann et al 2013 (lines 86-91); 92-103 a full description without any statistics; 104-114 RMSE 1-2cm in Dong and Menzel 2017 (please fix the citation in line 104). Finally, authors state "In conclusions, some works about the snow depth retrieval from time-lapse photography are available…". I would remove references and sentences without relevant details and I would find some comparison in the discussion section between the presented results and those available in literature.

Thank you for these comments. We fixed these in the new version of the manuscript. We tried to give relevant references regarding to importance of Snow Depth (SD), measurements of SD, usage of time-lapse photography on monitoring of snow and finally some work on retrieval of snow depth with accuracy parameters. We wanted to give information that webcams are already established in monitoring of snow in general not always specifically targeted certain parameters like SD. However we agree with the reviewer that too much details about references are given. We removed them in our new version.

Authors declare in the abstract (line 18-21) that "we present new possibilities and advantages in the retrieval of snow depth...which can be summarized as follows: 1) retrieving the snow depth at high temporal resolution with an accuracy comparable to the most common method (manual measurements); 2) visual identification of errors or misclassifications; 3) estimating the spatial variability of snow depth; 4) correcting the under catch problem of pluviometer when this instrument is used.

But they present the following results in the introduction (lines 124-126): "how the accuracy of snow depth estimations can be increased using stakes with 1 cm spacing markers; proper geometric and parameter configurations of the camera-stake system; sources of errors and post processing procedure corrections.

Thank you for these comments. In the abstract we reported all the advantages of this method, and how this method could be used in the future in the hydrological field highlighting which could be the

future applications. As you said, these applications, as estimating the spatial variability of snow depth and correcting the under-catch problem of pluviometer, aren't explained in details, so we removed these applications described advantages in the retrieval of snow depth as reported in the chapter 5 "Discussion".

**Methods**

Lines 267-271 the distance between the camera and the stake is still missing as well as the position in relation to the ultrasonic measurements.

Thank you for this comment. Yes, we fixed these data: Stick distance to the camera: 4.16 meters, Weather station - camera distance 222 meters.

Line 325 there is some text to be fixed close to the fig5 citation
Lines 325-334. Authors considered only the 2018-2019 period but describe a 2013-2109 starting interval adding also one figure (fig.2) that is wrongly numbered.
I suggest to mention only the considered period, the whole dataset and the considerations about problems for the visual observation create confusion to the reader.

Thank you for these comments. We fixed it in the revised version of the manuscript.

I suggest also to remove fig.5 since it is redundant with fig.11 and 12 where the same data are plotted again. I suggest also to merge fig. 11 and 12 evidencing the calibration and the validation periods.

Thank you for this comment. Yes, we agree to remove fig.5, the same information is available in fig.11 and 12. We thought that fig.11 and 12 must be split because:
1) Validation and calibration are two well different phases;
2) If we reported all the data series within only one figure, some details couldn't be clear and visible;

Line 358-371 I suggest to mention only the considered period, the whole dataset and the considerations about problems for the visual observation create confusion to the reader. I suggest also to remove fig.6 since it is redundant with fig.13 where the same data are plotted again. I suggest to evidence the calibration and the validation periods as for the Gressoney case study (there is no declaration in this case: 10 days as in Sodankyla?).

Thank you for this comment. Yes, we agree to remove fig.6, the same information is available in fig.13. About Sodankyla case study the calibration phase was done in the first observation year.

**Results**

Line 383 The distance between the camera and the stake is ideal but there is no info about that in the methods section.

Thank you for this comment. Yes we fixed these in the method section: Stick distance to the camera: 4.16 meters, Weather station - camera distance 222 meters.

Line 398 be careful declaring RMSE and NSE as accuracy estimations. Do Authors used the term accuracy differently in the introduction (precision, lowest limit of detection…..)?

No, we think that the term "accuracy" has to be use when we can define "a number" that explain how well our estimates reflect the available measurements.

Lines 402-404 no distances or maps have been provided to the reader.

We think that RMSE and NSE values explained well what we obtained and distances or maps are not necessary.

Lines 437-445 is the parallax effect only a local effect?

Yes, it's only a local effect. With careful positioning camera and stake we can easily remove this effect.

Lines 446-457 Why the melting near the stake is an issue only in this case study?

Not only in this case study, sometimes also in Sodankyla and Careser. This melting happened when the snow had a high liquid water content (especially in spring). Probably the weather in Careser (above 2600 m) and Sodankyla (Arctic Region), was characterized by less humidity, snow was dry for the most of the observation period.

How can you train your procedure without an AINEVA operator? This could be moved to the discussion section describing the difference between the AINEVA correction and the "standard" visual inspection.

AINEVA operators were the camera and stake's owner, so they gave us an estimation of the snow depth totally "independent" from our estimations. We thought that a comparison among AINEVA operators' estimations and our visual inspections wasn't the focus of this paper, because we had to define the method accuracy not the visual estimations one.

**Discussion**

There is a list of potential problems and of suggestions for the right configuration of observations. From my point of view, I would know:
1) what kind of corrections are still missing (parallax or heating-related issues for example). This could be firstly assessed looking at differences between visual observations on the stake and the AINEVA reading procedure.

Thank you for this comment. Parallax effect could be removed designing correctly the geometry and the position of camera and stake. The high melting rate depends on the snow condition, on the stake's material and it's hard to remove totally because varying among seasons.

2) what kind of differences are detected between ultrasonic measurements, photography-derived observations and other studies (Garvelmann et al 2013; Dong and Menzel 2017) that described similar measurements.

Ultrasonic measurements are very noisy in most of the case due to the sensor and its sensibility. Moreover, at the beginning of the snow accumulation season or the end of the melting season ultrasonic measurements could overestimate the snow because the grass growth. So, we think that coupling different estimation methods will be the right solution to better estimate the snow depth. Comparison with Gravelmann work it's hard because they didn't report any RMSE or NSE Value and they didn't compare the snow depth estimations with any kind of instrument, but just among different locations using the same method.

Dong and Menzel estimated the snow depth with a semi-automatic procedure based on time lapse photography, but only in one hydrological year, comparing its estimations with "manual" observation. So comparing these two works with our test site, not designed for time-lapse photography estimations could bring wrong interpretation on the method not due to the method itself but on the test sites.

Referring to the abstract statements, I don't see discussion about: retrieving the snow depth at high temporal resolution with an accuracy comparable to the most common method (manual measurements);

Thank you for this comment. The difference on the temporal resolution between manual measurements and time-lapse photography estimations, can be very large. In some cases, as in Careser Dam case study, manual measurements could be impossible to retrieve in winter because this site is at 2600 m above sea level with no road and connection.
In general, we can say that the manual measurement campaigns could be daily, if the site is supervised, but in most of the cases could be weekly or monthly. Time lapse photography method collects images potentially at the desired time resolution.

Estimating the spatial variability of snow depth; correcting the under catch problem of pluviometer when this instrument is used.
It is preferable to use the points cited in the introduction (lines 124-126): "how the accuracy of snow depth estimations can be increased using stakes with 1 cm spacing markers; proper geometric and parameter configurations of the camera-stake system; sources of errors and post processing procedure corrections.

Thank you for this comment. Yes "estimating the spatial variability of snow depth; correcting the under-catch problem of pluviometer" are possible future applications. The other improvements are already well described in the discussion section.

**General Comments**

We would like to thank the reviewer for very valuable comments. We modified and hopefully improved our manuscript in line with the reviewer's comment. We shortened introduction removing some details of state-of-art part from the introduction like:

"The inferred snow accumulation distribution was validated with in-situ measurements and correlations with topographic variables, such as curvature and slope, were found. So here, the focus is on a better representation of SWE and its spatial variability within a distributed hydrological model in mountain complex terrain and time-lapse photography is only a tool to improve results."

"For this purpose, they positioned two stakes painted with red color within the camera's view. They developed an algorithm that used pixel color intensities to automatically locate the snow surface for both the two stakes. The pixel counting algorithm clips each image to a small rectangle around the last known vertical pixel location of the marker base, then separates and smooths the blue channel of the RGB image for ultimate considerations. In fact, due to the high snow spectral albedo (that reflects most of the light in the near ultraviolet and blue visible spectrum), there was a large intensity gradient between depth marker and snow surface. This technique calculated a row wise RGB color minimum and the differences between two subsequent values. Snow surface was positioned near the maximum change in the reflectivity level. Finally, the snow depth retrieval was obtained by subtracting the pixel row from the snow free image, then dividing by the number of pixels per centimeter for the particular marker. The snow depth value was compared with ultrasonic snow depth sensor and one LIDAR survey, showing a high underestimation of the implemented algorithm for the period between December 2012 and April 2013. Any accuracy parameter was reported, but only a visual comparison was described."

"To estimate the snow depth, they used images in which a stake with red and black markers, each 10 cm, was visible. The procedure required three different software: Photoshop to extract a measure line of three pixels width from each snow stake in each picture. Then these measure lines were imported in ArcGIS and converted in ASCII files. Finally, the brightness series were further processed in Excel to extract the snow depth information. In particular they calculated the maximum brightness of the top 80 cm of the snow stake and adding a fixed value (20) they defined the threshold to detect the snow surface. A pixel was assumed to represent the snow surface when the brightness values below this pixel were higher than the previous mentioned threshold. The regression model of the automatically and manually interpreted snow depth values from the digital pictures were reported, which showed that the RMSE ranged from 0.011 to 0.019 m. Researchers should manually validate some abnormal values in accordance with the digital pictures obtained."

In addition, at the end of the introduction, i.e., in the two paragraphs starting with "In conclusion, some works about the snow depth retrieval ..." we have clarified our contribution respect to the existing literature, shaping better the main message which this work wants to vehiculate.

**Specific comments**

1) Eq. 1 in Introduction is likely not needed.

The SWE is the most common hydrological variable related to the snow, we think that it's important to remember the link between that variable and the snow depth. With this equation is much more clear why is important the Snow depth retrieval and how it's easy to link this variable with the SWE. Moreover another anonymous referee explicitly cited this formula saying:" the manuscript is focused on snow depth (looking at the title), good to say that SD is important for estimating SWE from eq.1".

2) L.55: "…but they were not specifically focused on the retrieval of snow depth." This is likely not a precise statement, as in the next sections, there is written that some studies (e.g. Parajka et al., 2012, Garvelman et . 2013, Dong et al., 2017) investigated the use of digital cameras for snow depth estimation.

We agree with the comment. It seems to be an apparential inconsistency. We removed the lines: "In these studies, the principal aim is to show how the time-lapse photography could be used for investigating snow processes, but they were not specifically focused on the retrieval of snow depth."

3) L.116: "there is no fully automatic procedure to retrieve the snow depth in real-time…". What is meant by fully automatic? In the next sentence, it is written as the new objective of this study, but the presented approach is not fully automatic, as you need to define several thresholds. So for an application in a new region, some setup and subjective decisions are needed.

Yes we agree with the reviewer that "a fully automatic procedure" can be understood differently. We meant "automated procedure" on retrieval steps on snow depth on defined ROI inputs. We replaced "a fully automatic procedure" with "an automated procedure" and we explained in the manuscript what it means :  In Line 117 "In this study we present an automated procedure with defined ROI inputs to retrieve the snow depth in real-time, and snow depth time series, from time-lapse photography. The automated procedure is based on an algorithm implemented within the FMIPROT, which uses the brightness difference between snowpack and stake's markers in digital images.

4) L.123: I would suggest turning the formulation "we presented.." into a formulation of the research hypothesis of this paper.

Ok. We fixed it through all manuscript.

5) L.185: I would suggest formulating a general procedure for a setup of the parameters in the Methods section. It should allow a reader/new user to understand and repeat the way/steps needed for estimating robust parameters in any region. Perhaps some objective calibration of these two parameters will be helpful. In its current form, it is not clear how many combinations and from which range the parameter selection should be based?

In Table 1 we reported the simulation's parameters. The threshold Ts can't vary widely in our case studies, from 67 to 70. The other parameter σ is much more sensitive, sometimes we used 1 in other case 10. But this variability, and it's dependence on the time window as in Careser Dam case study was due to the changing of the camera itself and we can not say that is due to the method inefficiency.

Will it be possible to "calibrate it" E.g. by minimising some RMSE or NSE??

Yes, it is possible to calibrate. But as all the methods are based on some threshold we have to collect a lot of images doing a sensitivity analysis of the result varying these thresholds. In this case we had not a long time series in which we could say that the geometrical configuration was the same, so this phase is just started.

Is this procedure sensitive to the selection of test window?

In our case, it is sensitive because the Careser camera changed the position.

6) Some figures are repeating the same observations (duplicating the same information).

Ok, we removed fig.5 and 6 because the same information (Snow depth time series in black dots) is reported in fig. 11 and 13.

 It will be interesting, perhaps, to see some additional evaluation of the sensitivity of the approach to the selection of the thresholds instead.

A true sensitivity analysis couldn't be done with a short time series of data.